# Keratinocytes mediate innocuous and noxious touch via ATP-P2X4 signaling

Francie Moehring[1], Ashley M Cowie[1], Anthony D Menzel[1], Andy D Weyer[1], Michael Grzybowski[2], Thiago Arzua[2], Aron M Geurts[2], Oleg Palygin[2], Cheryl L Stucky[1]*

[1]Department of Cell Biology, Neurobiology and Anatomy, Medical College of Wisconsin, Milwaukee, United States; [2]Department of Physiology, Medical College of Wisconsin, Milwaukee, United States

**Abstract** The first point of our body's contact with tactile stimuli (innocuous and noxious) is the epidermis, the outermost layer of skin that is largely composed of keratinocytes. Here, we sought to define the role that keratinocytes play in touch sensation in vivo and ex vivo. We show that optogenetic inhibition of keratinocytes decreases behavioral and cellular mechanosensitivity. These processes are inherently mediated by ATP signaling, as demonstrated by complementary cutaneous ATP release and degradation experiments. Specific deletion of P2X4 receptors in sensory neurons markedly decreases behavioral and primary afferent mechanical sensitivity, thus positioning keratinocyte-released ATP to sensory neuron P2X4 signaling as a critical component of baseline mammalian tactile sensation. These experiments lay a vital foundation for subsequent studies into the dysfunctional signaling that occurs in cutaneous pain and itch disorders, and ultimately, the development of novel topical therapeutics for these conditions.
DOI: https://doi.org/10.7554/eLife.31684.001

## Introduction

*For correspondence:
cstucky@mcw.edu

Competing interests: The authors declare that no competing interests exist.

Peripheral sensory neurons detect external stimuli and transmit this information to spinal cord and brainstem circuits. Despite their location below the epidermal surface, convention proposes that cutaneous sensory nerve terminals are the exclusive transducers of mechanical stimuli. This concept has recently been challenged by data that demonstrate epidermal Merkel cells' responsiveness to mechanical stimuli and subsequent signaling to sensory neurons, two processes that are essential for two-point touch discrimination (*Maksimovic et al., 2014*; *Woo et al., 2014*). Notably, Merkel cells constitute only a small portion (3–6%) of total skin cells (*Moll et al., 1986*; *Fradette et al., 2003*; *Halata et al., 2003*), whereas keratinocytes, which have traditionally been known for their roles in barrier formation and protection rather than sensory transduction, comprise 94–97% of the epidermis (*Fuchs, 1995*). However, keratinocytes are closely apposed to sensory nerve terminals (*Löken et al., 2009*) and are constantly exposed to external mechanical forces in the environment like brushing and pressure from stimuli like clothing, objects and other living organisms. Previously, it was demonstrated that isolated keratinocytes directly respond to mechanical probing by increasing intracellular calcium concentrations (*Koizumi et al., 2004*; *Tsutsumi et al., 2009*; *Goto et al., 2010*). Furthermore, keratinocytes can release many neuroactive substances including ATP, calcitonin gene-related peptide β (CGRPβ), acetylcholine, glutamate, epinephrine, neurotrophic growth factors, and cytokines among others (*Barr et al., 2013*; *Hou et al., 2011*; *Lumpkin and Caterina, 2007*; *Shi et al., 2013*). In co-cultures of keratinocytes and dorsal root ganglia (DRG) neurons, mechanical stimulation of keratinocytes evokes inward currents in adjacent sensory neurons, presumably through release of one of the aforementioned metabolites (*Klusch et al., 2013*). Taken together, these data suggest that sensory neurons may not be the sole transducers of mechanical

**eLife digest** The skin is the largest sensory organ of the body, and the first point of contact with the outside world. Whether it is being pinched or caressed, the skin's sense of touch informs organisms about their surroundings and allows them to react appropriately.

Nerve cells present in the skin capture information about touch and transmit it to the brain where it is decoded. However, there are many other types of cells in the skin besides nerve cells. The role that these other skin cells play in perceiving non-painful and painful touch is still unclear.

Moehring et al. now report how the skin cells that form 95% of the most outer layer of the skin are involved in detecting touch. In mutant mice whose cells can be 'switched off' by a certain light, artificially deactivating these cells makes the animals less able to respond to tactile stimuli. Further experiments show that when pressure is applied onto the skin, the surface skin cells release a chemical messenger, which then binds specifically to the nerve cells. When the messaging molecule is experimentally destroyed or prevented from attaching to the nerve cell, the mice react less to non-painful and painful touch. This means the cells at the surface of the skin detect tactile signals from the environment and then communicate this information to the nerve cells, where it is taken to the brain.

Disrupted communication between the cells in the outer layer of the skin and the nerve cells is found in painful and itchy skin conditions such as eczema and psoriasis. Knowing how these two types of cells normally work together may help with finding new pain and itch treatments for these skin disorders.

DOI: https://doi.org/10.7554/eLife.31684.002

stimuli, but rather may collaborate with other cell types such as keratinocytes to initiate or amplify somatosensory signals to sensory neurons.

Here, we sought to define the role of keratinocytes in mechanotransduction by utilizing cell-specific optogenetic approaches during evoked and non-evoked behavioral testing. We found that these epidermal cells are critical for innocuous and noxious touch detection. Using ex vivo sensory fiber recording techniques and pharmacological approaches, we identified ATP as a key signaling molecule released by keratinocytes in response to mechanical stimulation. Finally, we used novel sensory neuron-specific knockout mice to demonstrate that mechanically induced ATP release is functionally coupled to the activation of P2X4 receptors on sensory neurons. These data are the first to identify purinergic signaling as a critical component of innocuous and noxious skin mechanotransduction, specifically in the context of non-neuronal to neuronal cellular communication.

## Results

### Optogenetic inhibition of keratinocytes reduces mechanical responsiveness

We first sought to determine whether keratinocytes have a functional role in sensing touch. Keratinocytes were isolated from the glabrous hindpaw skin of transgenic mice that express *tdTomato* in *Keratin14* (*K14*)-positive (epidermal) cells, the vast majority (~94–97%) of which are keratinocytes (*Byrne et al., 1994*; *Dassule et al., 2000*; *Wang et al., 1997*), with a small percentage (3–6%) being Merkel cells (*Moll et al., 1986*; *Fradette et al., 2003*; *Halata et al., 2003*). Individual keratinocytes were visually identified and then subjected to focal mechanical stimulation or 'poke' (*Wu et al., 2017*) under current clamp conditions. Increasing indentation revealed a stimulus-dependent depolarization that returned to baseline between each stimulation (*Figure 1A*). We hypothesized that this depolarization may induce release of keratinocyte-derived factors that subsequently signal to sensory neurons, and therefore we aimed to utilize optogenetic approaches to manipulate this process.

A previous study that used optogenetic methods demonstrated that keratinocytes can modulate the responses of cutaneous sensory neurons in ex vivo skin nerve recordings (*Baumbauer et al., 2015*). However, this investigation stopped short of investigating the contributions of keratinocytes to tactile behavioral responses in vivo. Therefore, we created a mouse line that selectively expresses

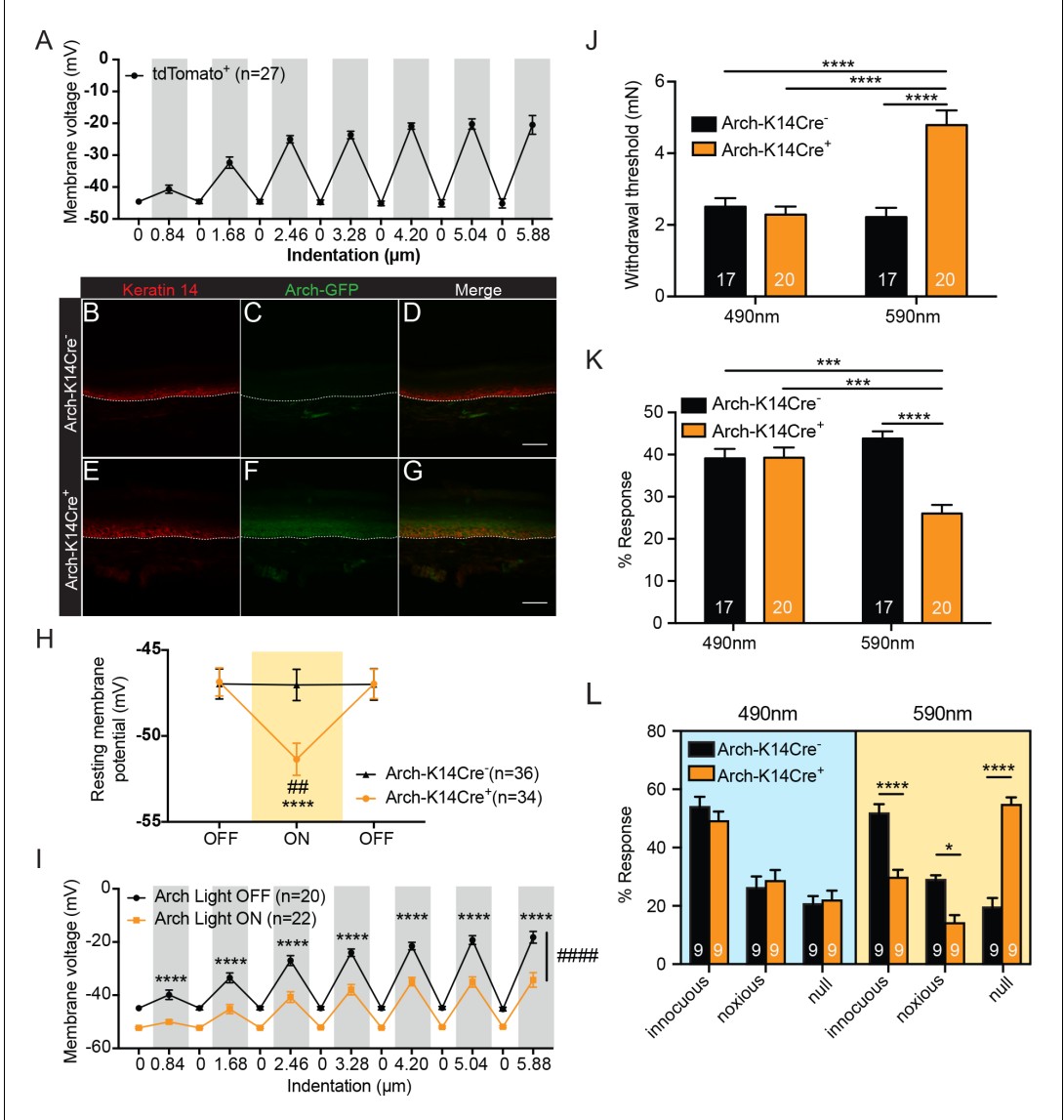

**Figure 1.** Keratinocytes depolarize upon mechanical stimulation, and optogenetic inhibition of keratinocytes decreases innocuous and noxious mechanical behavioral response. (**A**) In current clamp mode, mechanical stimulation of keratinocytes caused membrane depolarization of the soma membrane, grey background: mechanical stimulation, white background: resting membrane potential of the cells. (B-D Arch-K14Cre⁻) the dotted line indicates epidermal to dermal border. (**B**) Immunoreactivity of the K14 marker can be seen in the deeper layers of the skin (**C**) Archaerhodopsin is tagged with GFP and no significant GFP immunoreactivity is present in Arch-K14Cre- skin. (**D**) Overlay shows that the Arch-K14Cre⁻ animal has no GFP expression in K14-expressing cells. (E-G Arch-K14Cre⁺). (**E**) Immunoreactivity for K14 in red in the same areas as the Arch-K14Cre⁻ animal. (**F**) GFP expression is observed throughout the layers of the skin. (**G**) Merging of the red and green channels shows overlap of Archaerhodopsin-3 and the K14 marker in the Arch-K14Cre⁺ animal. Representative images are shown for each genotype for the immunohistochemistry experiments (n = 3 animals/ genotype), scale bar = 50 μm. (**H**) Arch-K14Cre⁺ keratinocytes had a more negative resting membrane potential when the LED light was turned on than when the light was off (****p<0.0001). Turning the LED (590 nm, 5 mW) on and off had no effect on the resting membrane potential of Arch-K14Cre⁻ keratinocytes (n.s.p=0.9937). In the light on condition Arch-K14Cre⁻ keratinocytes were significantly different from Arch-K14Cre⁺ keratinocytes (##p=0.0020), repeated measures two-way ANOVA, *Tukey* post-hoc test (n = 3–4 animals/genotype). (**I**) When the light (590 nm, 5 mW) was turned on in keratinocytes expressing Arch, the keratinocytes were hyperpolarized at baseline and showed an overall decrease (####p<0.0001) as well as a decrease at each indentation (0.84 μm:****p<0.0001, 1.68 μm:****p<0.0001, 2.46 μm:****p<0.0001, 3.28 μm:****p<0.0001, 4.20 μm:****p<0.0001 5.04 μm: ****p<0.0001 and 5.88 μm:****p<0.0001) in the membrane depolarization upon increasing indentations of the cell membrane, two-way ANOVA, *Sidak* post-hoc. (**J**) Von Frey Up-Down method showed that the 590 nm light significantly decreased normal baseline mechanical paw withdrawal thresholds in Arch-K14Cre⁺ animals in comparison to the Arch-K14Cre⁻ animals (****p<0.0001) as well as compared to the 490 nm control light (****p<0.0001). The 490 nm light had no effect on either genotype, two-way ANOVA, *Tukey* post-hoc. (**K**) Animals were stimulated 10 times with a suprathesheld 3.61 mN von Frey filament and the percent response was determined. Arch-K14Cre⁺ animals also showed fewer responses to the 3.61 mN stimulation when the

*Figure 1 continued on next page*

*Figure 1 continued*

590 nm light was on in comparison to the Arch-K14Cre⁻ controls (****p<0.0001) and the 490 nm light stimulation (***p<0.001) two-way ANOVA, *Tukey* post-hoc. (L) The hindpaw of animals was stimulated 10 times with a spinal needle and the responses were categorized into innocuous/normal response (simple withdrawal), noxious response (flicking, licking of the paw and elevating the paw for extended time periods) and null response. Arch-K14Cre⁺ mice showed fewer noxious (*p=0.0383), and innocuous (****p<0.0001), and concomitantly more null responses (****p<0.0001) to the needle stimulus, when exposed to the 590 nm light. There was no difference between genotypes in the type and number of responses when the 490 nm light was used (innocuous n.s. p=0.9957; noxious n.s. p>0.9999; null n.s. p>0.9999), three-way ANOVA, *Tukey* post-hoc. Throughout all the studies, the experimenter was blinded to genotype and treatment where possible.. Data are represented as mean ± SEM. See also *Figure 1—figure supplement 1*.

DOI: https://doi.org/10.7554/eLife.31684.003

The following figure supplement is available for figure 1:

**Figure supplement 1.** Light pre-treatment is not necessary to observe full behavior effects, and temperature increase in the skin due to fluorophore activation with the 590 nm LED is not responsible for the behavior responses observed in Arch-K14Cre⁺mice.

DOI: https://doi.org/10.7554/eLife.31684.004

GFP-tagged Archaerhodopsin-3 (Arch) in K14-expressing epidermal cells (*K14Cre⁺ Arch/Arch* (Arch-K14Cre⁺) and *K14Cre⁻ Arch/Arch* (Arch-K14Cre⁻) littermate controls) and tested whether keratinocytes have a functional role in sensing innocuous or noxious touch in vivo. When Arch is activated by amber light (peak photocurrent between 550 and 600 nm), it pumps protons out of the membrane, thereby hyperpolarizing the cell (*Chow et al., 2010*). Here, we activated Arch via transdermal light stimulation to inhibit epidermal cells in vivo.

To confirm that *Arch* expression was restricted primarily to epidermal cells, we evaluated GFP expression patterns in glabrous hindpaw skin sections. As expected, GFP (*Figure 1C,F*) overlapped substantially with K14-positive epidermal cells (*Figure 1B,E*) in Arch-K14Cre⁺ skin (*Figure 1G*), but not in Arch-K14Cre⁻ skin (*Figure 1D*). Because keratinocytes migrate from the basal to superficial epidermal layers in a temporal fashion, GFP expression was found throughout all layers, and was not restricted only to the basal keratinocyte layer where K14 expression is found.

We next assessed whether the Arch expressed in keratinocytes was functional. Whole cell current clamp recordings were performed on keratinocytes isolated from glabrous hindpaw skin of Arch-K14Cre⁺ and Arch-K14Cre⁻ mice in order to measure amber light (590 nm)-evoked changes in membrane potential. During light stimulation, Arch-K14Cre⁺ keratinocytes exhibited hyperpolarized membrane potentials, as compared to no light stimulation (*Figure 1H*). Light stimulation had no effect on the membrane potential of keratinocytes from Arch-K14Cre⁻ animals (*Figure 1H*). To determine whether optogenetic inhibition affects the mechanical responsiveness of keratinocytes, we recorded membrane voltage in Arch-K14Cre⁺ cells during focal stimulation and light exposure. Analogous to *Figure 1A*, the membrane voltage of Arch-K14Cre⁺ keratinocytes depolarized in a graded manner upon mechanical stimulation, and 590 nm light significantly reduced the overall level of depolarization at each force (*Figure 1I*). Hyperpolarization of keratinocytes significantly lowered the membrane potential even with mechanical stimulation, thus indicating that hyperpolarization can inhibit evoked signaling processes in keratinocytes.

To determine whether inhibition of K14-expressing cells affects animals' behavioral sensitivity to tactile stimuli, the glabrous hindpaw skin was briefly exposed to 590 nm light before (1 min) and during mechanical stimulation. Thresholds for tactile detection were measured using von Frey filaments (*Dixon, 1980*; *Chaplan et al., 1994*). Keratinocyte inhibition significantly elevated mechanical paw withdrawal thresholds in Arch-K14Cre⁺ animals compared to Arch-K14Cre⁻ controls (*Figure 1J*), reflecting decreased tactile sensitivity. Similar exposure to 490 nm light, a wavelength incapable of activating Arch, had no effect on mechanical thresholds (*Figure 1J*). Responses to repeated suprathreshold tactile stimuli were tested by applying a 3.61 mN filament 10 times to the plantar hindpaw and quantifying frequency (%) of withdrawal responses. The 590 nm light caused Arch-K14Cre⁺ animals to be less responsive to repeated probing than their Arch-K14Cre⁻ littermates (*Figure 1K*). Control light (490 nm) had no effect on the mechanical responsiveness of the Arch-K14Cre⁺ or Arch-K14Cre⁻ animals (*Figure 1K*). Although light pretreatments were given for 1 min before application of evoked stimuli for ease of stimuli administration, light pretreatment was not necessary to induce the behavioral mechanical inhibition, as 590 nm light treatment delivered simultaneously with the mechanical stimuli elicited the full effect of inhibition that was observed with pretreatment (compare *Figure 1—figure supplement 1A and B* to *Figure 1J and K*).

We next asked whether inhibition of K14-expressing cells also affects responses to noxious mechanical stimuli. The tip of a spinal needle was used to poke the hindpaw 10 times; responses were characterized as normal (innocuous, simple paw withdrawal), noxious (licking, flicking and elevating the paw for extended periods of time), or null (no response) (*Hogan et al., 2004*; *Moehring et al., 2016*). Arch-K14Cre$^+$ animals exposed to 590 nm light exhibited significantly fewer noxious and innocuous responses, and a concomitant increase in null responses to the needle poke compared to Arch-K14Cre$^-$controls (*Figure 1L*). Exposure to the control 490 nm light had no effect on the responses subtype distribution in either genotype (*Figure 1L*). Importantly, the behavioral changes observed in response to the 590 nm light were not a result of temperature changes in the hindpaw skin as neither 590 nm nor 490 nm light altered the temperature in the hindpaw skin (*Figure 1—figure supplement 1C and D*).

Next, to determine whether keratinocyte inhibition affects ongoing behaviors, the Arch-K14 cohorts were tested in a non-evoked place preference assay. Animals were allowed to freely explore a two-chamber box, where the floor of one chamber floor was illuminated with 595 nm light and the other was illuminated with 460 nm light. Neither Arch-K14Cre$^+$ nor Arch-K14Cre$^-$ animals preferred either chamber when the lights were on (*Figure 1—figure supplement 1E*), suggesting that the inhibition of keratinocytes and other epidermal cells alone does not evoke aversive or pleasant sensations in the animals. Together, these results demonstrate that epidermal K14-expressing cells play a key role in detecting evoked innocuous and noxious mechanical stimuli.

## Optogenetic activation of keratinocytes causes attending responses

Since light-induced inhibition of epidermal cells reduced the animals' baseline sensitivity to evoked mechanical stimuli, we performed complementary experiments to determine the effects of light-induced activation of K14-expressing epidermal cells using light-sensitive Channelrhodopsin 2 (ChR), which depolarizes cells when activated by 450 – 500 nm light (*Nagel et al., 2003*). We generated a mouse line that expresses eYFP-tagged ChR in K14-expressing cells (*K14Cre$^+$ ChR/ChR* (ChR-K14Cre$^+$) and *K14Cre$^-$ ChR/ChR* (ChR-K14Cre$^-$) littermate controls). eYFP expression was absent in ChR-K14Cre$^-$ skin (*Figure 2A–C*). Similar to the pattern of expression of *Arch* in Arch-K14Cre$^+$ sections, *ChR* expression was present (*Figure 2E*) throughout the keratinocyte layers, extensively overlapping with K14 immunoreactivity (*Figure 2D*) in ChR-K14Cre$^+$ animals (*Figure 2F*). To determine whether the ChR expressed in keratinocytes was functional, keratinocytes were isolated from the glabrous hindpaw skin of adult ChR-K14Cre$^+$ and ChR-K14Cre$^-$ mice and voltage clamped. All keratinocytes from ChR-K14Cre$^+$ animals responded with a sustained inward current during a 30 second 490 nm light stimulation (*Figure 2G–I*), whereas none of the ChR-K14Cre$^-$ keratinocytes responded (*Figure 2I*); only a small leak current was present in some cells. These data indicate that ChR is expressed and functional in adult mouse keratinocytes.

To determine if light-evoked activation of epidermal cells elicits behavioral responses, one hindpaw was focally exposed to 473 nm light. The time to initial response was measured and the type of response was categorized. On average, ChR-K14Cre$^+$ animals responded to hindpaw light stimulation within 46 seconds, whereas most ChR-K14Cre$^-$ controls did not respond to the 473 nm laser stimulation during the 6 min test (*Figure 2J*). The majority of mice from either genotype did not respond to the 589 nm control laser (*Figure 2—figure supplement 1A*). Additionally, ChR-K14Cre$^+$ animals spent significantly more time attending to their 473 nm light-stimulated hindpaw (*Figure 2—video 1*) than ChR-K14Cre$^-$ littermates (*Figure 2—video 2*); most of the ChR-K14Cre$^-$ mice were unresponsive to the stimulus for the 6 min duration of the test (*Figure 2—figure supplement 1B*). Once again, these results could not be accounted for by increases in skin temperature, as no difference was noted in skin temperature during light stimulation between genotypes and because either the 473 nm or 589 nm laser elicited the same small temperature increase in both genotypes (*Figure 2—figure supplement 1C and D*). Together, these data indicate that light-induced depolarization of K14-expressing cells elicits attending responses in vivo.

To determine if the light induced responses in K14-ChR animals displayed in *Figure 2J* and *Figure 2—figure supplement 1B* were aversive, we utilized a non-evoked real-time place preference assay (*Figure 2K*). ChR-K14Cre$^-$ and ChR-K14Cre$^+$ animals spent the same amount of time on the 595 nm and 460 nm-paired sides during light on and light off conditions (*Figure 2K*). Upon closer inspection of the animal behaviors on the two sides, it became apparent that the 460 nm light evoked significantly more 'grooming' and attending behaviors in the ChR-K14Cre$^+$ than in their

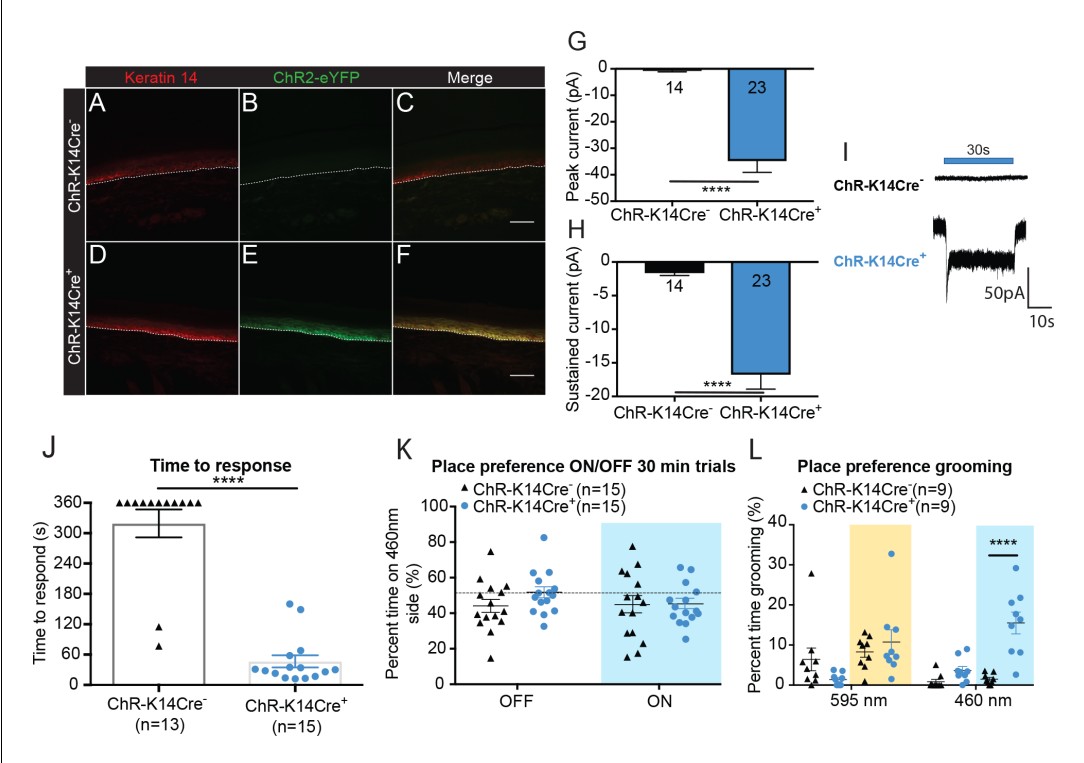

**Figure 2.** Optogenetic activation of keratinocytes elicits attending behavior responses. (A-C ChR-K14Cre⁻) the dotted line indicates epidermal to dermal border. (A) K14 immunoreactivity was observed in the deep layers of the skin. (B) ChR2 is tagged with eYFP; minimal eYFP fluorescence was observed in ChR-K14Cre- skin. (C) Merging shows no overlay of K14 and ChR2 in the ChR-K14Cre⁻ control animal. (D-F ChR-K14Cre⁺) (D) K14 immunoreactivity was observed in the deepest layers of keratinocytes. (E) eYFP fluorescence tag of the ChR2 expressing cells was observed all throughout the skin. (F) Merging of the red and green channels shows the overlap of the K14 marker with the eYFP expression of ChR2. Representative images are shown for each genotype for the immunohistochemistry experiments (n = 3 animals/genotype), scale bar = 50 μm. (G) Primary mouse keratinocytes were patch clamped in voltage clamp mode and stimulated for 30 seconds with a 3 mW 490 nm LED light (n = 4 animals/genotype). Keratinocytes cultured from ChR-K14Cre⁺ animals showed a significant increase in peak amplitude of the currents as compared to keratinocytes cultured from ChR-K14Cre⁻ animals which showed only a leak current (****p<0.0001) unpaired t-test. (H) The sustained current was significantly higher in ChR-K14Cre⁺ keratinocytes as compared to the ChR-K14Cre⁻ keratinocytes (****p<0.0001), unpaired t-test. (I) Example traces from ChR-K14Cre⁻ keratinocytes show no inward current (top), whereas ChR-K14Cre⁺ keratinocytes show an inward current in response to the light (bottom). (J) Animal behavior responses to a 473 nm 6 min laser stimulation (10 Hz, 24 – 28 mW power) were analyzed by an observer blinded to genotype. ChR-K14Cre⁺ animals responded in less than a minute to the laser stimulation, whereas most control animals did not respond at all to the laser stimulation (****p<0.0001) unpaired t-test. (K) K14-ChR animals were tested in a 30 min trial, optogenetic place preference set up. Neither genotype displayed a significant preference for either side during the 30 min on trial or the 10 min off trial (Light on vs off: n.s. p=0.3469; Genotype: n.s. p=0.3800), two-way ANOVA, *Sidak* post-hoc. (L) During the place preference assay, ChR-K14Cre⁺ animals spent significantly more time grooming on the 460 nm side when the light was on than their ChR-K14Cre⁻ controls (****p<0.0001). ChR-K14Cre⁺ animals on the 460 nm side with the light on also spent significantly more time grooming on that side than when the light was off (***p=0.0009); three-way ANOVA, *Tukey* post-hoc. Data are represented as mean ± SEM. See also *Figure 2—figure supplement 1* and *Figure 2—video 1* and *2*.

DOI: https://doi.org/10.7554/eLife.31684.005

The following video and figure supplement are available for figure 2:

**Figure supplement 1.** The 589 nm control laser does not elicit behavior responses, and attending behaviors are not due to heating of the skin.
DOI: https://doi.org/10.7554/eLife.31684.006

**Figure 2—video 1.** ChR-K14Cre⁺ animal responds to the 473 nm laser within 30 seconds of light stimulation.
DOI: https://doi.org/10.7554/eLife.31684.007

**Figure 2—video 2.** ChR-K14Cre⁻ animal does not respond to the 473 nm laser.
DOI: https://doi.org/10.7554/eLife.31684.008

littermate controls (*Figure 2L*). The aversive/grooming behaviors noted were repeated face 'wiping' with the forepaws, biting both fore- and hindpaws, and shaking of the tail. Taken together, these data make clear that keratinocyte signaling is necessary for naive behavioral responses to both innocuous and noxious mechanical stimuli, and that activation of keratinocytes and other epidermal cells alone is sufficient to elicit behavioral responses.

## Cutaneous mechanical stimulation induces ATP release from keratinocytes

Because we and others have shown the importance of keratinocyte signaling to sensory neurons (*Baumbauer et al., 2015*; *Pang et al., 2015*), we next sought to determine the signaling molecule(s) that mediate keratinocyte to sensory neuron communication. We investigated ATP because we recently showed that mice with a deficit in tactile sensitivity have decreased mechanically evoked ATP release from skin (*Zappia et al., 2016*). To assess levels of mechanically evoked ATP release from skin, ATP-sensing enzymatic probes were inserted into isolated glabrous hindpaw skin from naive wild-type mice (*Figure 3A*). The skin was then probed with forces ranging from 1.6 to 84.8 mN. Transient ATP release was detected during all of the epidermal stimulations (*Figure 3B*). To determine whether ATP release was graded according to stimulus intensity, increasing mechanical forces were applied to the glabrous skin with von Frey filaments. Increasing forces resulted in graded, increased ATP release (*Figure 3C*). Further, repeated stimulation with a single force (20.1 mN) elicited reproducible ATP release (*Figure 3D*). These data indicate that ATP is released in a reproducible and graded manner by mechanical stimulation of isolated glabrous hindpaw skin.

Next, we asked whether keratinocytes are the key source of ATP release in the skin. To do this, we performed a 'cell sniff' assay (*Lalo et al., 2014*; *2007*). HEK-293 cells were transfected with P2X2 receptors tagged with a C-terminal GFP. P2X2 transfected HEK-293 cells were co-cultured with K14Cre$^+$/tdTomato$^+$ keratinocytes obtained from adult mouse glabrous hindpaw skin. The P2X2 transfected HEK-293 cell-line (GFP$^+$) was used to detect ('sniff') ATP release during mechanical stimulation of a keratinocyte (tdTomato$^+$). P2X2-mediated currents in HEK-293 cells were monitored via whole cell voltage clamp while an adjacent keratinocyte was mechanically stimulated with a glass probe (*Figure 3E and F*). P2X2-GFP$^+$ HEK-293 cells, but not GFP$^-$ cells, showed robust inward currents when a nearby keratinocyte was mechanically stimulated as shown in current trace examples (*Figure 3G*) and current density (*Figure 3H*). Further, the magnitude of evoked current in GFP$^+$ HEK cells increased with increasing indentation of the nearby keratinocyte (*Figure 3H*). These results demonstrate that keratinocytes from naive adult mouse skin are capable of rapidly releasing ATP in response to mechanical probing and that the amount of ATP released is indentation-dependent.

We next asked whether optogenetic manipulations of isolated keratinocytes could alter mechanically evoked ATP release. We used the same cell sniff assay as in *Figure 3H*, except that the P2X2-GFP$^+$ HEK-293 cells were co-cultured with Arch-K14-Cre$^+$ keratinocytes. Keratinocytes were mechanically stimulated in the presence or absence of 590 nm light, and inward currents in the P2X2-GFP$^+$ HEK-293 cells were recorded. Exposure to the 590 nm light blunted the amplitude of the mechanically evoked currents in the sniffer cells compared to the light off current amplitudes (*Figure 3I*). In addition to optogenetic inhibition, we also tested the converse experiment by co-culturing ChR-K14Cre$^+$ keratinocytes with P2X2-GFP$^+$ HEK-293 cells. To determine if optogenetic-induced depolarization was sufficient to cause inward currents in the P2X2-GFP$^+$ HEK-293 cells, 490 nm light with one of three different intensities (0.2, 2 or 20 mW), or a 590 nm (5 mW) control light, was shone on the ChR-K14Cre$^+$ keratinocytes while inward currents in the P2X2-GFP$^+$ HEK-293 cells were measured. Light-induced 490 nm depolarization was sufficient to elicit inward currents in an intensity-dependent manner, whereas no depolarization occurred with 590 nm light (*Figure 3J*). Collectively, these data indicate that ATP release from keratinocytes is largely voltage-dependent and that its release is amenable to optogenetic manipulations

## ATP hydrolysis decreases acute mechanical responsiveness

We next asked whether ATP release in skin is required for normal behavioral responses to tactile stimuli. Apyrase, an enzyme that catalyzes ATP hydrolysis (*Palygin et al., 2015*, *Palygin et al., 2017*), was injected subcutaneously into the hindpaw and sensitivity to tactile stimuli was tested. Apyrase injection significantly elevated paw withdrawal thresholds compared with vehicle-injected

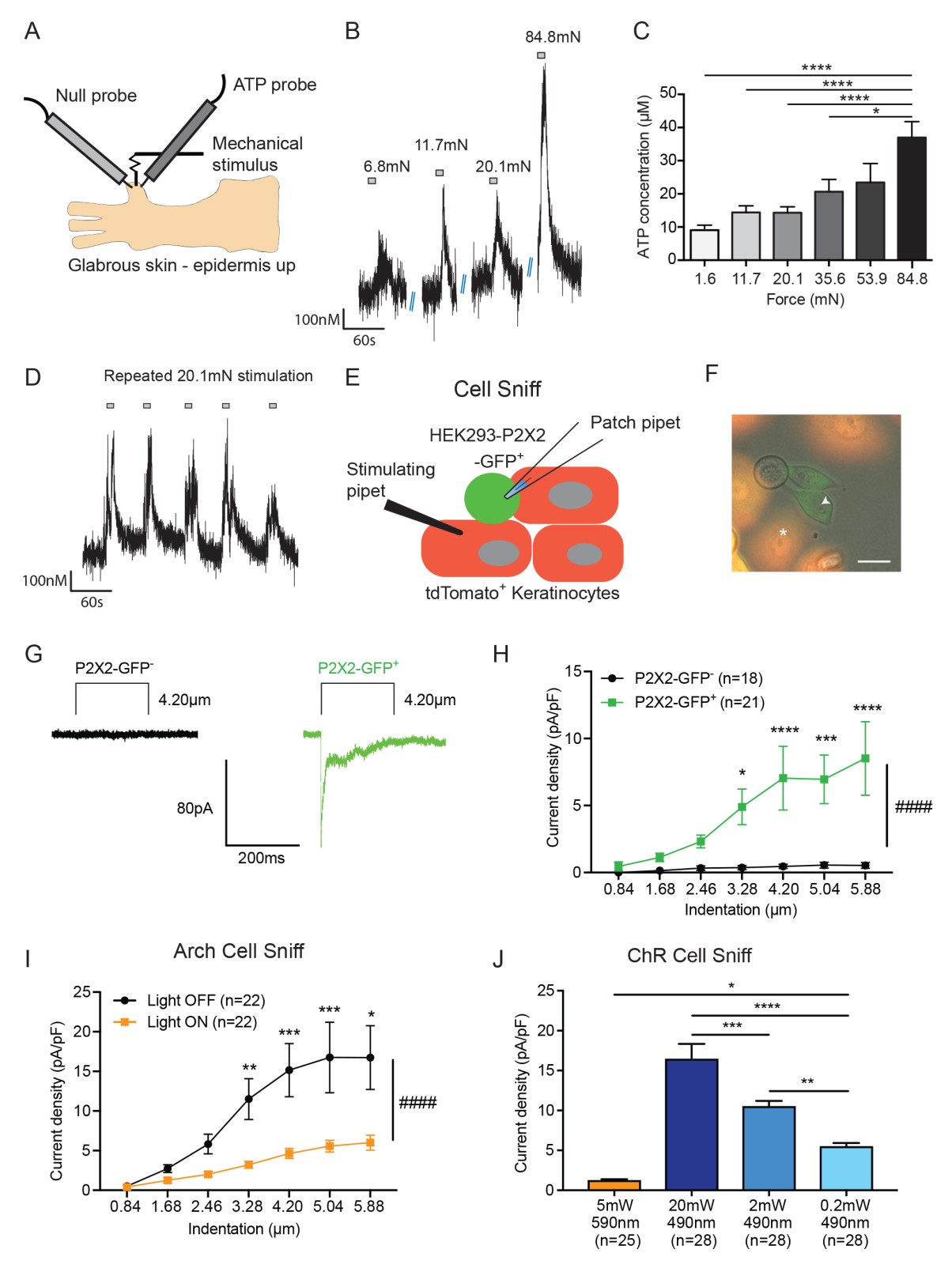

**Figure 3.** ATP is released from both mechanically- and optogenetically-stimulated keratinocytes, and optogenetic inhibition of keratinocytes dampens this ATP release. (**A**) Schematic of ex vivo glabrous skin setup with ATP and null probes inserted into the skin. (**B**) Repeatable ATP release traces are shown upon skin stimulation with 6.8, 11.7, 20.1, and 84.8 mN von Frey filaments; for each filament, the skin was rapidly and repeatedly stimulated with the von Frey Filament for 10 seconds. (**C**) The area under the curve for ATP release was quantified. ATP release was significantly greater in
*Figure 3 continued on next page*

Figure 3 continued

response to the 84.8 mN stimulus than the 35.6 mN (*p=0.0220), 20.1 mN (****p<0.0001), 11.7 mN (****p<0.0001) and 1.6 mN stimuli (****p<0.0001) (n = 4 animals) one-way ANOVA, *Tukey* post-hoc test. (D) Traces of ATP release upon repeated mechanical stimulation with the same 20.1 mN force show approximately the same amount of ATP release for each stimulus. (E) Schematic of the cell sniff assay. P2X2 GFP HEK-293 cells (sniffer cells) were patch clamped and primary cultured keratinocytes were mechanically stimulated with a stimulating pipet with increasing increments. (F) Bright-field image merged with the red and green fluorescent channel shows P2X2-GFP[+]HEK-293 cells with K14-tdTomato-tagged keratinocytes. Star = keratinocyte that is mechanically stimulated; arrow head = P2X2-GFP[+] HEK-293 cell that is patch clamped, scale bar 20 μm. (G) Current traces showing example currents of P2X2-GFP[+] HEK-293 cells and P2X2 GFP[-] HEK-293 cells in response to mechanical probing (4.20 μm) of a keratinocyte. (H) P2X2-GFP[+] HEK-293 cells show a gradual and significant increase in current density in response to increasing mechanical stimulation of the keratinocyte, which is not observed in P2X2 GFP[-] HEK-293 cells (####p<0.0001), (GFP[-] vs GFP[+]: 4.20 μm ***p=0.0009; 5.04 μm **p=0.0034; 5.88 μm ***p=0.0003; 6.72 μm ****p<0.0001; 7.56 μm ****p<0.0001) two-way ANOVA, *Sidak's* post-hoc test. (I) Arch inhibition of K14-expressing cells during mechanical stimulation significantly decreased the current density in response to mechanical stimulation in P2X2-GFP[+] HEK-293 cells (####p<0.0001); (Light OFF vs Light ON: 3.28 μm **p=0.0044; 4.20 μm ***p=0.0003; 5.04 μm ***p=0.0006; 5.88 μm *p=0.0233) two-way ANOVA, *Sidak's* post-hoc test. (J) In the ChR cell sniff assay, the control light (590 nm) did not elicit significant currents in P2X2-GFP[+] HEK-293 cells. Conversely, 490 nm light-induced depolarization of ChR keratinocytes via three different light intensities was sufficient to cause inward currents in P2X2-GFP[+] HEK-293 cells, compared to the 590 nm light control (5 mW 590 nm vs 20 mW 490 nm: ****p<0.0001, 5 mW 590 nm vs 2 mW 490 nm: ****p<0.0001 and 5 mW 590 nm vs 0.2 mW 490 nm: *p=0.0249). The magnitude of response in the P2X2-GFP[+] HEK-293 was light intensity dependent, where a greater light intensity elicited a greater current (20 mW 490 nm vs. 2 mW 490 nm: ***p=0.0004; 20 mW 490 nm vs. 0.2 mW 490 nm: ****p<0.0001 and 2 mW 490 nm vs. 0.2 mW 490 nm **p=0.0040) one-way ANOVA, *Tukey* post-hoc test. Data are represented as mean ± SEM.

DOI: https://doi.org/10.7554/eLife.31684.009

animals (*Figure 4A*). Apyrase-treated animals also responded significantly fewer times to repeated suprathreshold stimulation than vehicle-treated animals (*Figure 4B*). Further, Apyrase-injected animals showed a significant decrease in both the number of noxious and innocuous responses to needle stimulation and a concomitant increase in the number of null responses (*Figure 4C*). Therefore, acute hydrolysis of ATP in hindpaw skin reduces the responsiveness to both noxious and innocuous mechanical stimuli under baseline conditions. Importantly, investigation of apyrase at the cellular level through utilizing the cell sniff assay demonstrated that the apyrase used in this study does indeed degrade ATP, as the inward currents in P2X2-GFP[+] HEK-293 cells were attenuated when apyrase was added to the extracellular buffer while nearby keratinocytes were mechanically stimulated evoked (*Figure 4D*).

Since in vivo hydrolysis of peripheral ATP reduced behavioral tactile and noxious mechanical sensitivity, we next asked whether mechanically evoked action potential firing of peripheral sensory neurons depends on cutaneous ATP release. To explore this question, we used the tibial nerve ex vivo preparation, which mirrors the anatomical location of mechanical stimuli applied in the behavioral assays. We first recorded from C-fibers because C-fiber terminals are closely apposed to keratinocytes and non-peptidergic C-fibers project most superficially in the epidermis (*Zylka et al., 2005*). The isolated glabrous skin of the tibial nerve preparation was exposed to apyrase or PBS in the bath. Overall, C-fibers treated with apyrase fired significantly fewer action potentials in response to a series of increasing forces than those treated with PBS (*Figure 4E*). Examples of C-fiber action potentials in the presence of apyrase or PBS are shown in *Figure 4F*. Apyrase had no effect on the mechanical thresholds as evaluated by von Frey filaments (PBS: 5.88 ± 5.88 mN, apyrase: 5.88 ± 5.88 mN; median ± interquartile range; Mann-Whitney U-test, p=0.60) or conduction velocity (PBS: 0.55 ± 0.22 m/s, apyrase: 0.53 ± 0.24 m/s; mean ± SEM; unpaired t-test, p=0.79) of C-fibers. We next tested myelinated slowly adapting (SA) Aδ and Aβ-fibers, which are typically found slightly deeper within the keratinocyte layers of the epidermis. Apyrase significantly reduced the number of mechanically evoked action potentials fired in both SA-Aδ and SA-Aβ-fibers (*Figure 4G and I*). Example traces of Aδ and Aβ-fiber action potentials show that apyrase decreases the number of action potentials fired in response to increasing forces (*Figure 4H and J*). Like C-fibers, apyrase had no effect on the mechanical thresholds of Aδ-fibers (PBS: 4.00 ± 2.82 mN, apyrase: 4.00 ± 2.82 mN; median ± interquartile range; Mann-Whitney U-test, p=0.70) or Aβ-fibers (PBS: 4.00 ± 2.82 mN, apyrase: 4.00 ± 5.19 mN; median ± interquartile range; Mann-Whitney U-test, p=0.80) as measured by von Frey filaments, or the conduction velocities of Aδ-fibers (PBS: 7.28 ± 0.47 m/s, apyrase: 6.60 ± 0.55 m/s; mean ± SEM; unpaired t-test, p=0.36) or Aβ-fibers (PBS: 17.71 ± 2.87 m/s, apyrase: 14.31 ± 0.81 m/s; mean ± SEM; unpaired t-test, p=0.25). Furthermore, in order to ensure that apyrase would not directly alter membrane excitability of sensory neurons, lumbar dorsal root ganglia (DRG)

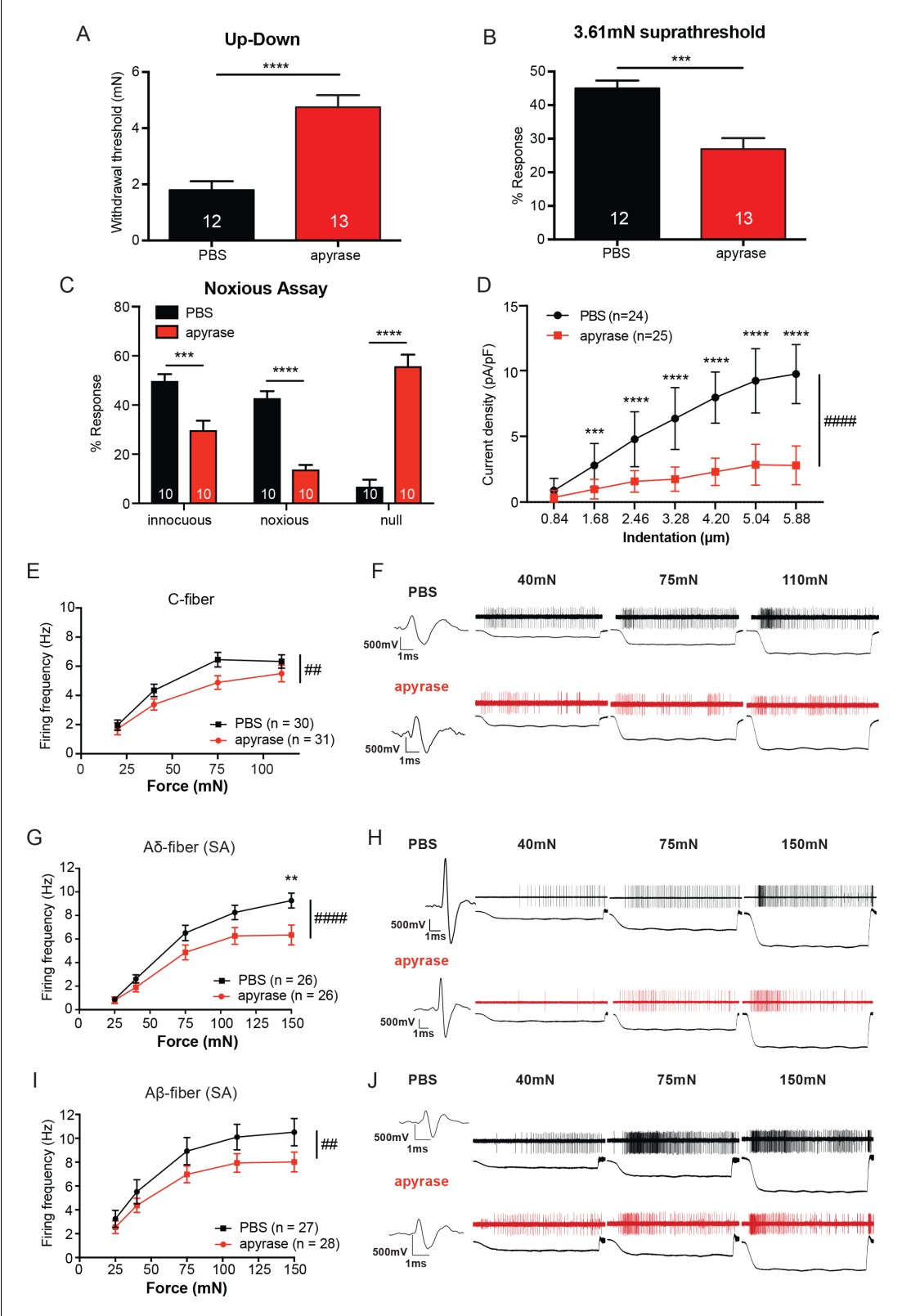

**Figure 4.** The degradation of ATP leads to decreased innocuous and noxious behavioral touch responses and decreased afferent firing of all fiber types. (**A**) Animals injected with 0.4 units apyrase had a two-fold increase of the paw withdrawal threshold as compared to their vehicle controls (****p<0.0001), Mann-Whitney U-test. (**B**) Animals injected with apyrase had significantly lower percent responses than animals injected with the vehicle (***p=0.0004), Mann-Whitney U-test. (**C**) Apyrase treatment decreased both the innocuous (***p=0.0004) and noxious (****p<0.0001) responses and

*Figure 4 continued on next page*

*Figure 4 continued*

simultaneously increased the percent of null responses (****p<0.0001) as compared to the PBS treated animals, two-way ANOVA, *Tukey* post-hoc. (D) Keratinocytes co-cultured with P2X2-GFP[+] HEK-293 cells were incubated with 20 units of apyrase or PBS. The presence of apyrase decreased the current density in response to mechanical stimulation of keratinocytes in P2X2-GFP[+] HEK-293 cells (####p<0.0001); (PBS vs apyrase: 1.68 μm ***p=0.0007; 2.46 ****p<0.0001; 3.28 μm ****p<0.0001; 4.20 μm ****p<0.0001, 5.04 μm ****p<0.0001; 5.88 μm ****p<0.0001) two-way ANOVA, *Sidak's* post-hoc test. (E) Ex vivo tibial skin nerve preparations were incubated with either 40 units of apyrase or PBS, for at least 10 min prior to any recordings. C-fibers were characterized based on their conduction velocity of <1.2 m/s. C-fiber action potentials in response to a force ramp of 20, 40, 75 and 110 mN over a 12 second period were recorded. Apyrase treatment overall significantly decreased the action potential firing rate of the C-fibers (##p=0.0051), two-way ANOVA, *Sidak* post-hoc test (n = 22 mice). (F) C-fiber traces on top with PBS and on the bottom with apyrase. (G) Slowly Adapting (SA) Aδ-fibers were characterized based on their conduction velocity of 1.2–10 m/s and repeated firing to sustained stimuli. Action potentials in response to a force ramp of 25, 40, 75, 110 and 150 mN over a 12 second period were recorded. Apyrase treatment overall decreased action potential firing over the different forces as compared to PBS treatment in the Aδ-Fibers (####p<0.0001), and at the 150 mN force, apyrase action potential firing was significantly decreased as compared to PBS (**p=0.0017), two-way ANOVA, *Sidak* post-hoc test. (H) SA-Aδ-fiber action potential traces, on the top with PBS treatment, bottom with apyrase treatment. (I) SA-Aβ-fibers were characterized based on their conduction velocities > 10 m/s and repeted firing to sustained stimuli. There was a significant decrease in action potential firing in the SA-Aβ-fibers treated with apyrase as compared to those incubated with PBS (##p=0.0025), two-way ANOVA, *Sidak* post -hoc test. (J) SA-Aβ-fiber action potential traces, on the top with PBS treatment, bottom with apyrase treatment. (n = 42 mice for Aδ and Aβ-fibers). Throughout all the studies, the experimenter was blinded to the treatment. Data are represented as mean ± SEM. See also *Figure 4—figure supplement 1*.

DOI: https://doi.org/10.7554/eLife.31684.010

The following figure supplement is available for figure 4:

**Figure supplement 1.** Apyrase treatment does not affect sensory neuron membrane excitability.
DOI: https://doi.org/10.7554/eLife.31684.011

neurons from naive animals were isolated and current clamped in the presence of a high concentration of apyrase or vehicle. Apyrase treatment did not alter the rheobase (current required to elicit one action potential) values as compared to the PBS control (*Figure 4—figure supplement 1A*). In addition, apyrase treatment did not alter the resting membrane potential of the neurons (*Figure 4—figure supplement 1B*). Together, these data demonstrate that cutaneous ATP signaling is essential for normal behavioral responses to innocuous and noxious mechanical stimuli and is required at cutaneous terminals for afferent firing elicited by mechanical stimuli.

## Degradation of ATP mirrors behavioral responses during optogenetic inhibition of K14-expressing cells

We have thus far discovered that: (1) inhibition of keratinocytes decreases innocuous and noxious touch responses, (2) mechanical stimulation of keratinocytes releases ATP, and (3) degradation of ATP in skin decreases mechanical sensitivity at the behavioral and afferent levels. We next asked whether keratinocytes are the major source of ATP released from the skin upon mechanical stimulation. The combined apyrase treatment and 590 nm light inhibition of K14-expressing epidermal cells had no additive effect on behavioral mechanical thresholds (*Figure 5A*), or on responses to repeated suprathreshold stimulation (*Figure 5B*) when compared to vehicle with 590 nm light. This suggests that K14-expressing cells are the major source of ATP in skin. Importantly, the mechanical thresholds and suprathreshold responses in apyrase-treated animals (in every genotype and light condition) were not different from the Arch-K14Cre[+] PBS treated 590 nm light condition (*Figure 5A and B*). As expected, apyrase treatment alone markedly decreased the animals' mechanical sensitivity as is evident by the increased paw withdrawal thresholds (*Figure 5A*) and decreased responses in the suprathreshold assay (*Figure 5B*). Further, the non-specific 490 nm light had no effect in either Arch-K14Cre[+] or Arch-K14Cre[-] cohorts (*Figure 5A and B*). In the noxious needle assay, the effects of apyrase treatment were not significantly different from that of both optogenetic inhibition and apyrase treatment, again suggesting that K14-expressing cells are a major source of the ATP that is required for noxious mechanosensation (*Figure 5C*). The control 490 nm light had no effect on either Arch-K14Cre[+] or Arch-K14Cre[-] cohorts (*Figure 5—figure supplement 1A*). Taken together, these findings indicate that other non-K14-expressing cells are not a significant source of mechanically evoked ATP release in skin. Furthermore, because there was no additive effect of apyrase together with optogenetic inhibition of keratinocytes, the data make clear that ATP is the major signaling molecule released from keratinocytes in response to innocuous and noxious mechanical stimulation of normal skin.

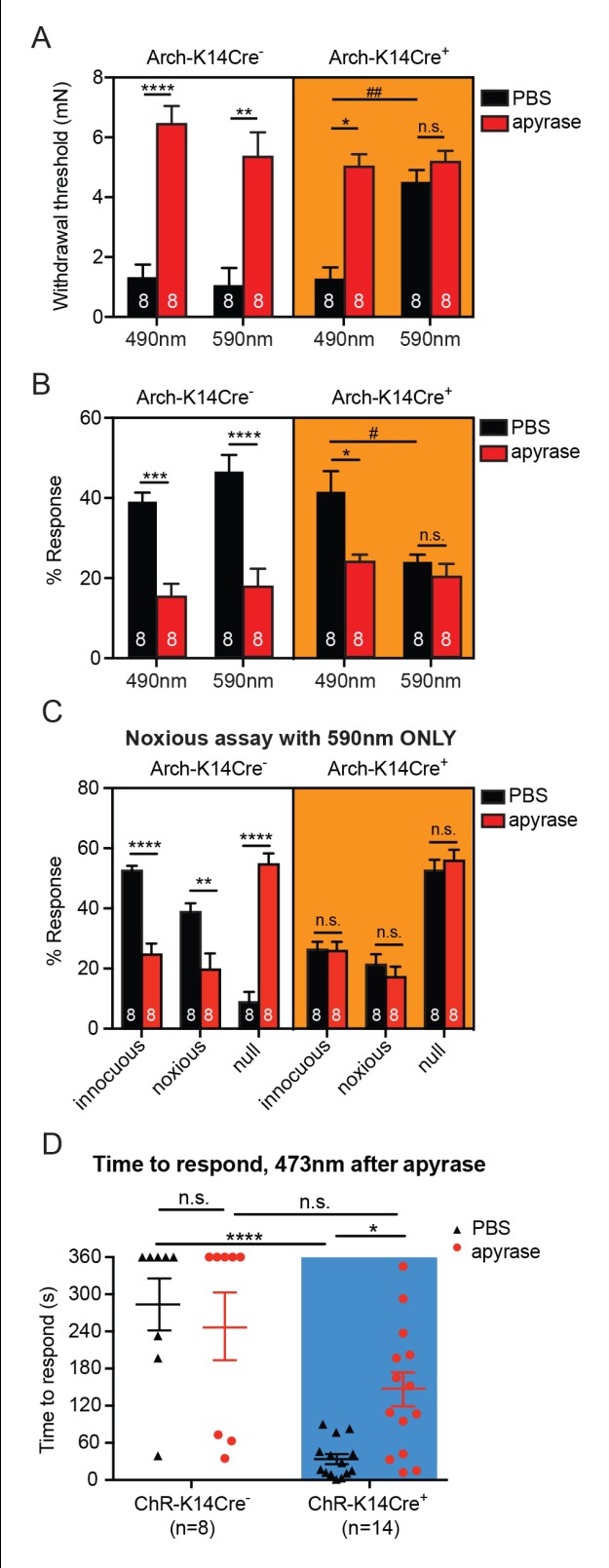

**Figure 5.** Degradation of ATP decreases touch sensitivity and responsiveness to a similar level as that with keratinocyte inhibition, and delays behaviors elicited by keratinocyte-activation.. (**A**) Animals were tested 45 min after apyrase/vehicle injection with either the 490 or 590 nm light on. Arch-K14Cre[+] PBS animals showed significantly higher paw withdrawal thresholds with the 590 nm light on than with the 490 nm light on

*Figure 5 continued on next page*

*Figure 5 continued*

(##p=0.0093), and Arch-K14Cre$^+$ animals treated with apyrase had significantly higher paw withdrawal thresholds than Arch-K14Cre$^+$ animals treated with PBS (*p=0.0168), three-way ANOVA, *Tukey* post-hoc. (B) Arch-K14Cre$^+$ animals treated with apyrase were not significantly different from the 590 nm light stimulation and PBS or apyrase treatment. Arch-K14Cre$^+$ treated with apyrase had approximately half of the responses compared to Arch-K14Cre$^+$ animals treated with PBS under the 490 nm light (*p=0.0175). Apyrase-treated animals did not differ from each other between genotypes or light wavelength; three-way ANOVA, *Tukey* post-hoc. (C) Graph shows 590 nm light treatment only. After apyrase treatment, Arch-K14Cre$^+$ animals treated with apyrase were no longer significantly different from animals treated with PBS (innocuous: n.s. p>0.9999; noxious: n.s. p=0.9996; null: n.s. p=0.9996). Conversely, apyrase- treated Arch-K14Cre$^-$ animals exhibited significantly more null responses (****p<0.0001) and less innocuous (****p<0.0001) and noxious (**p=0.0064) responses when compared to their PBS-treated littermates, three-way ANOVA, *Tukey* post-hoc. (D) ChR-K14Cre$^+$ animals treated with PBS responded significantly faster to the laser light stimulation than ChR-K14Cre$^+$ animals injected with apyrase (*p=0.0368). ChR-K14Cre$^+$ animals injected with PBS responded significantly sooner to the laser stimulation that ChR-K14Cre$^-$ animals injected with PBS (****p<0.0001). ChR-K14Cre$^+$ animals injected with apyrase no longer differed from the ChR-K14Cre$^-$ controls injected with apyrase (n.s. p=0.1241), two-way ANOVA, *Tukey* post-hoc. Throughout all the studies, the experimenter was blinded to genotype as well as treatment. Data are represented as mean ± SEM. See also *Figure 5—figure supplement 1*.
DOI: https://doi.org/10.7554/eLife.31684.012
The following figure supplement is available for figure 5:

**Figure supplement 1.** Apyrase treatment in control behavior experiments does not differ between genotypes.
DOI: https://doi.org/10.7554/eLife.31684.013

---

Next, we assessed whether the light-evoked increase in behavioral responses in ChR-K14Cre$^+$ mice was specifically due to ATP release from keratinocytes by injecting apyrase into hindpaw skin. Apyrase-treated ChR-K14Cre$^+$ animals exhibited their first response to the 473 nm light significantly later than the ChR-K14Cre$^+$ animals treated with PBS (*Figure 5D*). Furthermore, the response times of apyrase-treated ChR-K14Cre$^+$ mice were similar to those of either apyrase- or PBS-treated ChR-K14Cre$^-$ animals (*Figure 5D*). When animals did respond to the 473 nm light, apyrase treatment had no effect on the type of response in either ChR-K14Cre$^+$ or ChR-K14Cre$^-$ animals (*Figure 5—figure supplement 1B*). Exposure to 589 nm laser stimulation failed to initiate a response in either genotype or treatment group (*Figure 5—figure supplement 1C*). These data indicate that light-induced activation of keratinocytes is sufficient to evoke attending behavioral responses and that ATP release from skin is an essential signaling molecule involved in these attending responses. Taken together, these results show that ATP is a major signaling molecule released from keratinocytes especially in response to mechanical stimulation.

## ATP released from keratinocytes acts on P2X4 receptors on sensory neurons

Next, we asked which receptor on sensory nerve terminals responds to the ATP released from keratinocytes. Although there are many P2X family members, P2X4 receptors were of particular interest due to their high abundance and relatively equal expression in C and A-fiber neurons (*Kobayashi et al., 2013*). To pharmacologically inhibit P2X4 in the periphery, the selective P2X4 inhibitor 5-BDBD (5-(3-Bromophenyl)-1,3-dihydro-2H-benzofuro[3,2-e]-1,4-diazepin-2-one) was injected into one plantar hindpaw and mechanical sensitivity was tested. 5-BDBD significantly increased the mechanical thresholds in naive animals (*Figure 6A*) and significantly decreased the responsiveness to repeated mechanical probing compared to vehicle (*Figure 6B*). Because P2X2 and P2X3 have been shown to be involved in various pain states (*Novakovic et al., 1999*; *Cockayne et al., 2000*; *North, 2004*; *Bernier et al., 2017*) and because they are also highly expressed on sensory neurons (*Kobayashi et al., 2013*) we tested if P2X2 and P2X3 receptors could play a role in baseline mechanical sensation. To pharmacologically inhibit P2X3 and P2X2/3 receptors, two concentrations of NF 110 were injected subcutaneously into the plantar hindpaw of naive animals. At a low concentration (500 nM), NF 110 inhibits P2X3, but at a high concentration (5 mM), NF 110 inhibits both P2X2 and P2X3 receptors (*Hausmann et al., 2006*). Neither concentration affected baseline mechanical sensitivity as measured in either the Up-Down mechanical threshold or in the suprathreshold assay, in these mice 60 min after subcutaneous injection (*Figure 6C and D*).

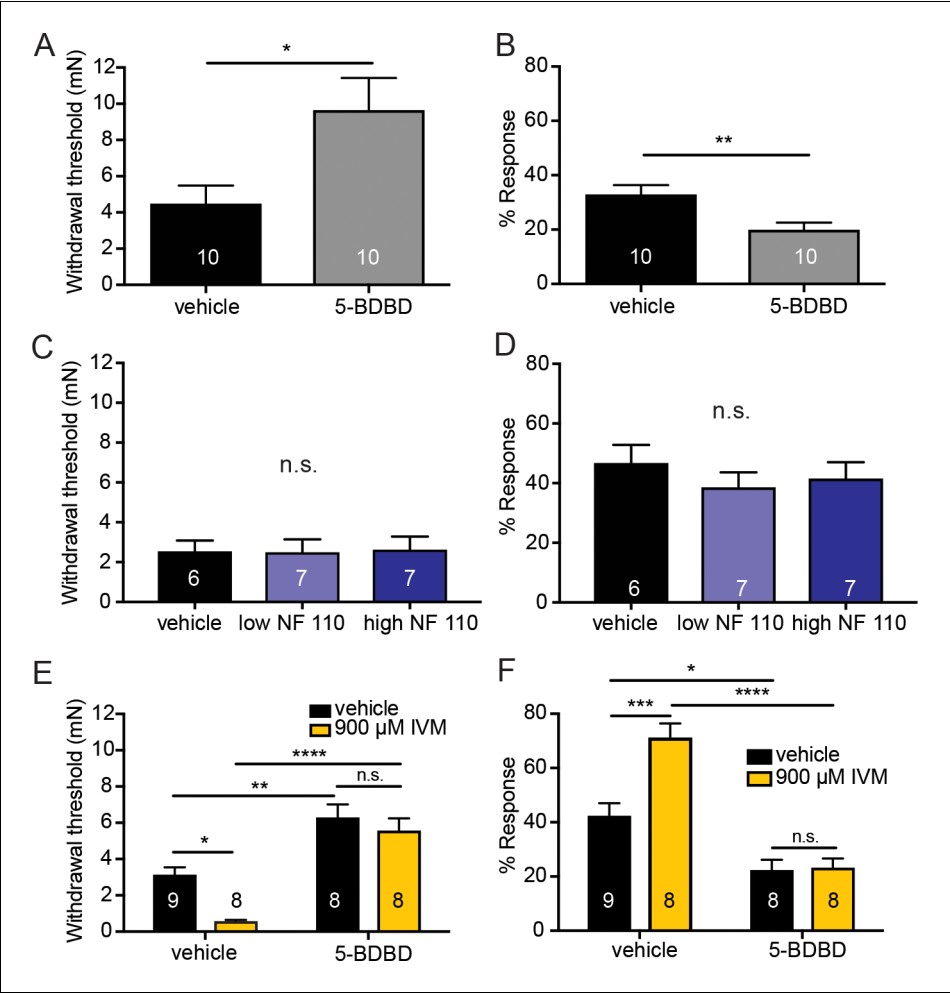

**Figure 6.** Pharmacological inhibition of peripheral P2X4 and not P2X2 and P2X3 receptors causes decreased mechanical responses in vivo. (**A**) Naive C57BL/6J mice were injected with either 5-BDBD (selective P2X4 antagonist) or vehicle, 1 hr prior to behavioral experiments. Animals injected with 5-BDBD had significantly higher paw withdrawal thresholds than their vehicle controls (*p=0.0190), Mann-Whitney U-test. (**B**) Animals injected with 5-BDBD showed fewer responses to the 3.61 mN stimulus than the vehicle control (**p=0.0033), Mann-Whitney U-test. (**C**) Naive C57BL/6J mice were injected with either vehicle, low NF 110 (500 nM) or high NF 110 (5 mM) 1 hr prior to behavior assays. Neither the low nor the high concentration of NF 110 had an effect on the paw withdrawal thresholds of the animals (n.s. p=0.9946), Kruskal-Wallis test. (**D**) Injection of low- or high concentration NF 110 had no effect on the percent response to a suprathreshold 3.61 mN stimulus (n.s. p=0.6083), Kruskal-Wallis test. (**E**) 5-BDBD or vehicle was injected 1 hr prior to the behavior assays and 20 min prior to the assays animals were also injected with 900 µM ivermectin or vehicle. Ivermectin with vehicle injection decreased the paw withdrawal thresholds significantly as compared to the ivermectin and 5-BDBD combined injection (****p<0.0001), two-way ANOVA, *Tukey* post-hoc. (**F**) Ivermectin injection with 5-BDBD caused significantly fewer responses than when ivermectin was injected with the vehicle control (****p<0.0001), two-way ANOVA, *Tukey* post-hoc. Furthermore, ivermectin with 5-BDBD was not significantly different from the 5-BDBD with vehicle control (n. s. p=0.9989), two-way ANOVA, *Tukey* post-hoc. Data are represented as mean ± SEM. The experimenter was blinded to compound treatment. See also *Figure 6—figure supplement 1*.
DOI: https://doi.org/10.7554/eLife.31684.014

The following figure supplement is available for figure 6:

**Figure supplement 1.** P2X4 inhibition (5-BDBD) or potentiation (ivermectin) does not affect membrane excitability in sensory neurons.
DOI: https://doi.org/10.7554/eLife.31684.015

These data demonstrate that mechanically induced release of ATP from keratinocytes is most likely acting through P2X4 receptors.

Conversely, to determine if P2X4 activation sensitizes animals to mechanical stimulation, ivermectin, an anti-parasitic drug that potentiates P2X4 currents induced by ATP binding (*Priel and Silberberg, 2004*), was used. Subcutaneous ivermectin administration significantly decreased mechanical withdrawal thresholds (*Figure 6E*) and significantly increased responsivity to repeated suprathreshold stimulation (*Figure 6F*). Pretreatment with the P2X4 inhibitor 5-BDBD prevented the ivermectin-induced mechanical sensitization in both the mechanical threshold (*Figure 6E*) and repeated force assays (*Figure 6F*), indicating that the effects of ivermectin were due to P2X4 activation. Once again, these behavioral results could not be accounted for by altered membrane excitability of sensory neurons due to high concentrations of 5-BDBD or ivermectin as neither drug altered the rheobase (*Figure 6—figure supplement 1A and C*) or resting membrane potential of sensory neurons (*Figure 6—figure supplement 1B and D*) compared to their vehicle controls.

To determine whether P2X4 receptor activation was specific to receptors on cutaneous sensory nerve terminals, we generated mice with a selective deletion of *P2rx4* in sensory neurons. Mice expressing Cre under the control of the sensory neuron-*Advillin* promoter (*da Silva et al., 2011*; *Zappia et al., 2017*) were crossed with mice carrying a *P2rx4 loxP* conditional knockout allele (*Yang et al., 2014*) to produce *AdvillinCre*+*P2X4*fl/fl*(P2X4-AdvCre*+*) and *AdvillinCre*-*P2X4*fl/fl* (P2X4-AdvCre*-) littermate controls. (*Yang et al., 2014*). To confirm that *P2rx4* is decreased in DRG, we performed qRT-PCR on whole DRG homogenates. As expected, there was a significant decrease in *P2rx4* mRNA in lumbar DRG of P2X4-AdvCre+ animals (*Figure 7—figure supplement 1*). P2X4 sensory neuron mutants (P2X4-AdvCre+) displayed significantly higher paw withdrawal thresholds (*Figure 7A*) and significantly fewer responses to repeated mechanical probing when compared with control littermates (*Figure 7B*). P2X4-AdvCre+ animals were also less responsive to noxious mechanical stimulation. These animals had significantly fewer noxious and innocuous responses and more null responses during the noxious needle assay than littermate controls (*Figure 7C*). Furthermore, ivermectin injection had no effect on mechanical withdrawal thresholds (*Figure 7D*) or responsiveness to repeated suprathreshold stimulation (*Figure 7E*) in P2X4-AdvCre+ animals.

Finally, to determine whether other P2X or P2Y channels besides P2X4 were activated by the mechanically released ATP, hindpaws of sensory neuron P2X4 mutants were injected with apyrase or PBS. P2X4-AdvCre- animals injected with apyrase had significantly higher paw withdrawal thresholds than P2X4-AdvCre- animals injected with PBS (*Figure 7F*). However, apyrase treatment had no additional effect on mechanical withdrawal thresholds in P2X4-AdvCre+ animals; apyrase treatment had the same effect in both P2X4 expressing and P2X4 sensory neuron mutant mice (*Figure 7F*). Similarly, apyrase had no effect on P2X4-AdvCre+ animal responsiveness to repeated suprathreshold stimulations (*Figure 7G*) or the noxious needle assay (*Figure 7H*), but did reduce the responses in P2X4-AdvCre- littermate controls (*Figure 7G and H*). In conclusion, these data demonstrate that either pharmacological inhibition or genetic deletion of P2X4 channels specifically from sensory nerve terminals is sufficient to fully decrease the baseline mechanical sensitivity to both innocuous and noxious force.

Since the *Advillin*-driven knockout of P2X4 reduced innocuous and noxious mechanical sensitivity, we next asked whether mechanically evoked action potential firing of peripheral sensory neurons was also affected. Single nerve recordings from the ex vivo tibial nerve preparation revealed that C-fibers from P2X4-AdvCre+ animals had overall significantly reduced firing rates as compared to P2X4-AdvCre- controls (*Figure 8A and B*). Furthermore, the action potential firing rate of slowly adapting myelinated Aδ and Aβ-fibers in response to increasing mechanical stimuli was also decreased overall (*Figure 8C and E*). Example traces for Aδ and Aβ-fibers are shown in *Figure 8D and F*. In contrast to the apyrase studies where mechanical thresholds of fibers were measured via traditional von Frey filaments, here we utilize a newly designed custom feedback-controlled mechanical stimulator which applied a force ramp of 0 to 100 mN to determine mechanical thresholds. Remarkably, C-fiber, Aδ-fiber and Aβ-fibers from P2X4-AdvCre+ animals had significantly higher action potential thresholds than their wild type littermates (*Figure 8G–I*). Of note, we have never before found such a major shift in von Frey threshold at the single fiber level for any knockout or injury model in skin nerve recordings by using von Frey filaments. These data indicate that P2X4 on sensory neuron terminals participates in setting the threshold for mechanical firing of multiple classes of cutaneous sensory neurons. Conduction velocities were not altered by the knockdown of

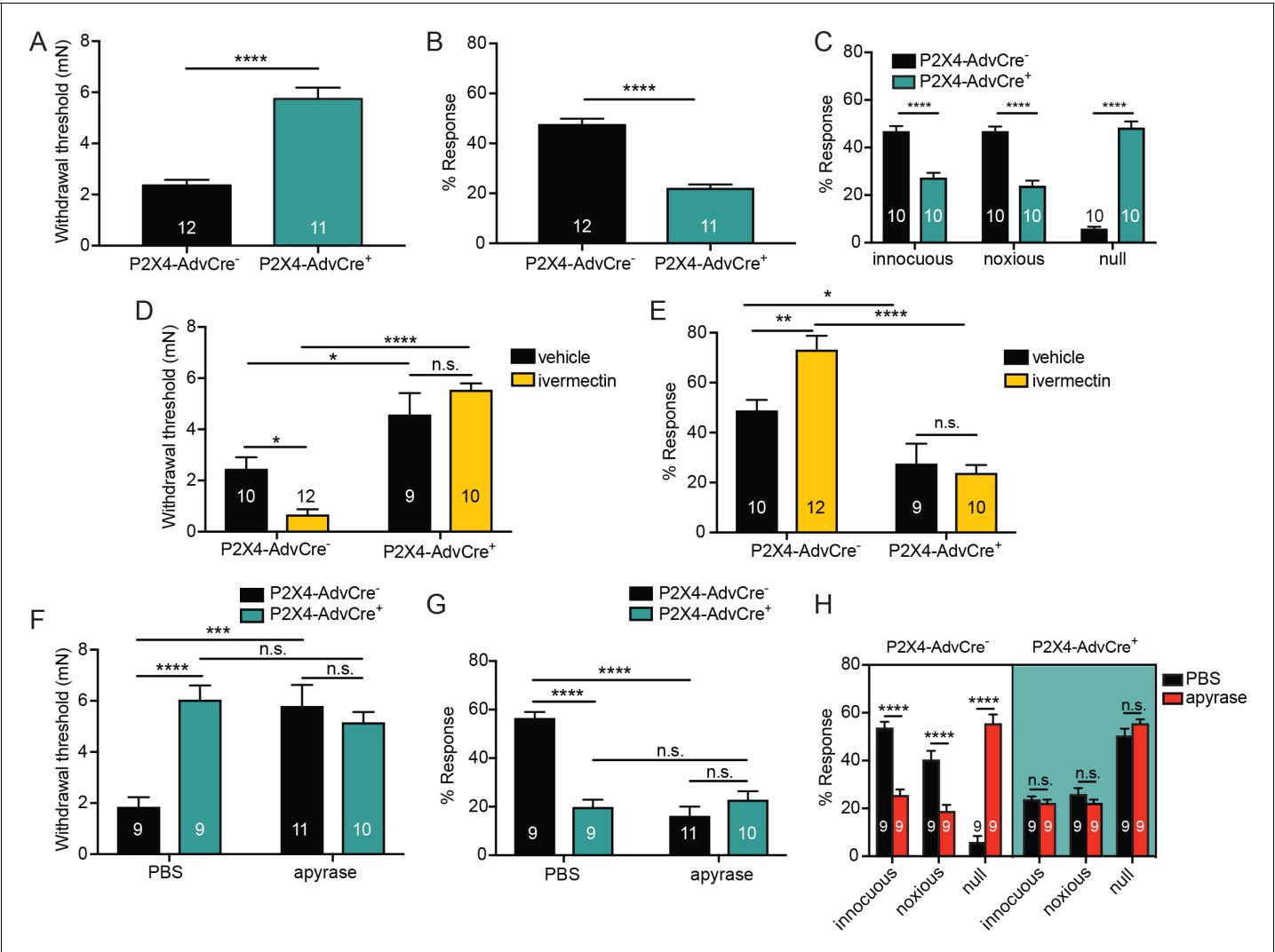

**Figure 7.** ATP is primarily acting through P2X4 receptors on sensory neurons to mediate innocuous and noxious touch in vivo. (**A**) P2X4-AdvCre⁺ animals had higher paw withdrawal thresholds than their littermate controls (****p<0.0001), Mann-Whitney U-test. (**B**) P2X4-AdvCre⁺ animals also showed fewer responses to the 3.61 mN stimulus than their controls (****p<0.0001), Mann-Whitney U-test. (**C**) P2X4-AdvCre⁺ animals showed a significant increase in null responses in the noxious needle assay (****p<0.0001) as compared to P2X4 AdvCre⁻ animals. P2X4-AdvCre⁺ animals also showed a significant decrease in innocuous (****p<0.0001) and noxious responses (****p<0.0001) as compared to their controls; two-way ANOVA, *Tukey* post-hoc. (**D**) P2X4-AdvCre⁺ animals and littermates were injected with 900 µM ivermectin 20 min prior to the behavior assays. P2X4-AdvCre⁺ showed no sensitization after ivermectin injection (n.s. p=0.4605), whereas P2X4-AdvCre⁻ controls with ivermectin injection had significantly lower paw withdrawal thresholds compared to the vehicle injected P2X4-AdvCre⁻ animals (*p=0.0273) and the ivermectin injected P2X4-AdvCre⁺ animals (****p<0.0001), two-way ANOVA, *Tukey* post-hoc. (**E**) P2X4-AdvCre⁺ animals injected with ivermectin were not sensitized after ivermectin injection (n.s. p=0.9615) and had significantly fewer responses than their littermate controls injected with ivermectin (****p<0.0001); two-way ANOVA, *Tukey* post-hoc. (**F**) Apyrase treatment significantly elevated the paw withdrawal threshold of P2X4 AdvCre⁻ animals (****p<0.0001), but it had no additional effect on P2X4-AdvCre⁺ animals (n.s. p=0.6573), two-way ANOVA, *Tukey* post-hoc. (**G**) Apyrase treatment significantly lowered the number of responses to the 3.61 mN filament in P2X4-AdvCre⁻ animals (****p<0.0001), but it had no additional effect on P2X4-AdvCre⁺ animals (n.s. p=0.9126). Furthermore, P2X4-AdvCre⁻ apyrase treated animals were not significantly different from P2X4-AdvCre⁺ apyrase treated animals (n.s. p=0.4330), two-way ANOVA, *Tukey* post hoc. (**H**) Apyrase treatment in P2X4-AdvCre⁻ animals significantly elevated the null responses (****p<0.0001) and decreased both the innocuous (****p<0.0001) and noxious (****p<0.0001) responses as compared to P2X4-AdvCre⁻ animals treated with PBS. Apyrase treatment had no additional effect on P2X4-AdvCre⁺ animals compared to the PBS treatment (innocuous n.s. p>0.9999; noxious n.s. p=0.9993; null n.s. p=0.9543), three-way ANOVA, *Tukey* post-hoc. Data are represented as mean ± SEM. The experimenter was blinded to genotype and compound treatment. See also *Figure 7—figure supplement 1*.

DOI: https://doi.org/10.7554/eLife.31684.016

The following figure supplement is available for figure 7:

**Figure supplement 1.** P2X4-AdvCre⁺ animals have a ~70% knockdown of *P2rx4* in sensory neurons.

*Figure 7 continued on next page*

*Figure 7 continued*

DOI: https://doi.org/10.7554/eLife.31684.017

P2X4 in any fiber type; C-fibers (P2X4-AdvCre⁻: 0.72 ± 0.28 m/s, P2X4-AdvCre⁺: 0.76 ± 0.27 m/s mean ± SEM; unpaired t-test p=0.565), Aδ-fibers (P2X4-AdvCre⁻: 3.64 ± 1.91 m/s, P2X4-AdvCre⁺: 3.88 ± 2.29 m/s; mean ± SEM; unpaired t-test p=0.661) or Aβ-fibers (P2X4-AdvCre⁻: 12.87 ± 2.67 m/s, P2X4-AdvCre⁺: 12.76 ± 2.98 m/s; mean ± SEM; unpaired t-tests, p=0.913). Together, these data identify sensory neuron P2X4 as a key target of the ATP released from keratinocytes in response to both noxious and innocuous touch and show that sensory neuron-expressed P2X4 is essential for setting both the threshold for initiating mechanical firing and for regulating the firing frequency to suprathreshold sustained stimuli in several classes of slowly adapting sensory neurons in the skin.

## Discussion

Innocuous and noxious touch impacts our daily life, activities, and communication with other people and animals on a moment-to-moment basis. For such a ubiquitous phenomenon, it is surprising that so little is known about the mechanism underlying how these signals are conveyed from skin impact to the brain. Conventional theories of touch biology indicate that sensory neuron terminals are the initial and exclusive responders to harmless and painful touch. However, here we offer a paradigm shift to this dogma of somatosensory mechanotransduction. We show that keratinocytes, which cover the entire body and are the first cells to contact physical stimuli, are indispensable for normal innocuous and noxious touch sensation. Mechanistically, keratinocytes communicate with sensory nerve terminals via ATP release, which then activates P2X4 receptors on sensory neurons to signal both innocuous and painful touch perception from the skin (*Figure 9*).

### Keratinocytes are essential for detecting gentle and painful touch in normal skin

Our study builds on the elegant ex vivo findings of Bambauer and colleagues (2015) to demonstrate the tactile function of keratinocytes in the awake, behaving animal. Specific optogenetic inhibition of epidermal cells in Arch-K14Cre⁺ mice elevated the behavioral tactile thresholds, dampened the responses to suprathreshold innocuous force, and blunted the responses to noxious pin prick, indicating that keratinocytes have a major role in conveying a broad range of innocuous and noxious mechanical information to the CNS. Although this optogenetic silencing probably inhibited Merkel cells, which also express *Keratin14*, the contribution from this cell type to the overall behavioral observations is likely small since Merkel cells comprise only a very small portion (3–6%) of epidermal cells located in the glabrous or hairy skin (*Moll et al., 1986*; *Fradette et al., 2003*; *Halata et al., 2003*). Therefore, the K14 epidermal cell inhibition in vivo is largely mediated by keratinocytes.

### Keratinocyte activation elicits attending and 'grooming' behaviors

Complementary experiments, where keratinocytes were selectively activated by Channelrhodopsin in ChR-K14Cre⁺ mice, showed that specific keratinocyte activation caused animals to attend or 'groom' in response to the light-illuminated body regions by wiping of the face or front paws in both the evoked behavior and non-evoked place preference assay. While both assays reveal attending and grooming-like behaviors, there are also major differences in the types of behaviors observed. The light-evoked behavior from focal hindpaw stimulation appears to be nocifensive (*Pang et al., 2015*), with behaviors mimicking those observed in experiments utilizing optogenetic activation of TRPV1⁺ sensory neurons, which causes nocifensive behaviors as well as place aversion (*Beaudry et al., 2017*). However, in our hands, the place-preference assay did not cause a place aversion in the ChR-K14Cre⁺ animals, and the attending behaviors elicited by the 460 nm floor light are most reminiscent of paresthesia-like behaviors (*Kahn Safdar et al., 2012*). Further experiments must be done to determine the repertoire of sensations elicited during these interesting behaviors. Since the same set of epidermal cells are activated by 460 nm light in both assays, we believe the discrepancies in these two behavioral assays are due to different light power intensities and consequently, different levels of keratinocyte depolarization (75.2 μW LED floor in the place preference

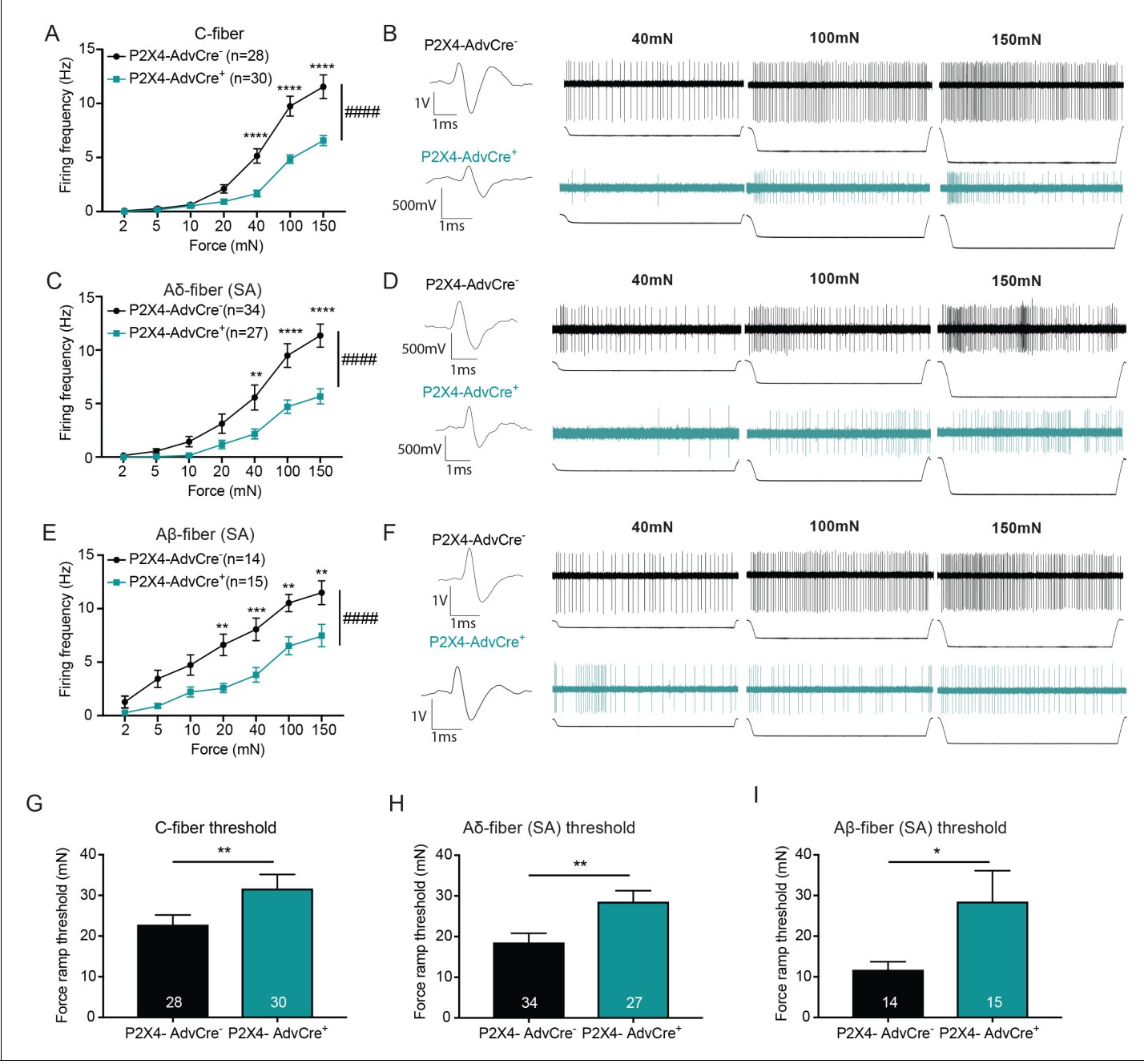

**Figure 8.** The knockdown of P2X4 in *Advillin*-expressing neurons significantly decreases afferent firing and elevates mechanical action potential thresholds of slowly adapting (SA) C-fibers, SA-Aδ and SA-Aβ-fibers. Recordings were made from tibial skin-nerve preparations of P2X4-AdvCre⁻ and P2X4-AdvCre⁺ animals. Action potentials were recorded with a force ramp of 2, 5, 10, 20, 40, 100 and 150 mN over a 10 second period. (A) C-fibers were characterized based on their conduction velocity of <1.2 m/s. P2X4-AdvCre⁺ C-fibers overall fired significantly fewer action potentials (####p<0.0001), (40 mN ****p<0.0001; 100 mN ****p<0.0001; 150 mN ****p<0.0001), two-way ANOVA, *Sidak* post-hoc test (n = 23 mice). (B) C-fiber traces, top: P2X4-AdvCre⁻ and bottom: P2X4-AdvCre⁺. (C) Aδ-fibers were characterized based on their conduction velocity of 1.2–10 m/s and repeated firing to sustained force. P2X4-AdvCre⁺ SA-Aδ-fibers overall fired significantly fewer action potentials (####p<0.0001), (40 mN **p=0.0055; 100 mN ****p<0.0001; 150 mN ****p<0.0001), two-way ANOVA, *Sidak* post-hoc test (n = 23 mice). (D) SA-Aδ-fiber action potential traces, top: P2X4-AdvCre⁻ and bottom: P2X4-AdvCre⁺. (E) Aβ-fibers were characterized based on their conduction velocities > 10 m/s and repeated firing to sustained force. P2X4-AdvCre⁺ SA-Aβ-fibers overall fired fewer action potentials (####p<0.0001), (20 mN **p=0.0020; 40 mN ***p=0.0010; 100 mN **p=0.0023; 150 mN **p=0.0023), two-way ANOVA, *Sidak* post-hoc test (n = 23 mice). (F) SA-Aβ fiber action potential traces, top: P2X4-AdvCre⁻ and bottom: P2X4-AdvCre⁺. G) Action potential firing thresholds of the different fiber subtypes were determined using a force ramp from 0 to 100 mN over a 10 second period. C-fiber thresholds were significantly elevated in P2X4-AdvCre⁺ preparations as compared to the P2X4-AdvCre⁻ preparations (*p=0.0446), unpaired t-test. (H) SA-Aδ-fiber action potential thresholds in P2X4-AdvCre⁺ mice were significantly higher than in littermate controls

*Figure 8 continued on next page*

*Figure 8 continued*

(**p=0.0065), unpaired t-test. (I) P2X4-AdvCre⁺ SA-Aβ-fibers action potential thresholds were significantly higher than those of P2X4-AdvCre⁻ SA-Aβ-fibers (*p=0.0497), unpaired t-test. Data are represented as mean ± SEM.

DOI: https://doi.org/10.7554/eLife.31684.018

assay vs. 25 mW laser in the hindpaw light-evoked assay, a 333-fold difference in light intensity). This hypothesis is further supported by findings in our ChR cell sniff assay where P2X2 expressing HEK-293 cells exhibit intensity-dependent increases in inward currents in response to light stimulation of ChR-expressing keratinocytes, indicating an intensity-dependent release of ATP by keratinocytes. These data show that keratinocyte depolarization alone is sufficient to cause attending behaviors in freely moving animals.

## Native keratinocytes depolarize in response to mechanical stimuli

It may be surprising that keratinocyte function in vivo can be modulated by optogenetic manipulation of the membrane because these cells do not fire action potentials. However, we show here that primary keratinocytes from adult glabrous skin do depolarize upon mechanical stimulation in an indentation-dependent manner. Moreover, Arch inhibition during focal mechanical stimulation of keratinocytes significantly reduced the overall level of depolarization at each indentation. Conversely, we show that ChR activation directly elicits inward currents in P2X2 expressing HEK-293 cells, thereby showing that the depolarization of keratinocytes is sufficient to release ATP. Other reports using keratinocyte cell lines show that keratinocytes can depolarize or hyperpolarize in response to changes in extracellular ionic gradients (*Wohlrab et al., 2000*), exhibit increased intracellular $Ca^{2+}$ in response to mechanical stimuli (*Koizumi et al., 2004*; *Tsutsumi et al., 2009*; *Goto et al., 2010*), and express voltage-gated sodium and calcium channels, as well as Transient Receptor Potential channels (*Denda et al., 2006*; *Zhao et al., 2008*; *Caterina and Pang, 2016*). These prior reports suggest that keratinocytes possess the functional ion channels required for producing rapid changes in membrane excitability, and our new data reveal that naive adult mouse keratinocytes depolarize in response to mechanical stimuli.

## Mechanical stimulation of the skin and keratinocytes elicits ATP release

We next sought to determine how keratinocytes must be communicating with sensory nerve terminals. This process likely occurs via a chemical signaling pathway based on evidence of synapse-like structures between keratinocytes and sensory nerve terminals (*Hilliges et al., 1995*; *Chateau and Misery, 2004*; *Château et al., 2007*; *Klusch et al., 2013*; *Roggenkamp et al., 2013*) and the fact that keratinocytes have been shown to contain and release a variety of neurotransmitter molecules (*Burrell et al., 2005*; *Lumpkin and Caterina, 2007*; *Barr et al., 2013*; *Hou et al., 2011Shi et al., 2013*). We found at the cellular, tissue and behavioral levels evidence that ATP is released from keratinocytes in response to mechanical stimulation of the skin. Experiments utilizing ex vivo glabrous hindpaw skin show that mechanical stimulation elicits reproducible and graded ATP release, and experiments employing a cell sniff assay verify that keratinocytes are specifically responsible for the ATP release, which occurs in an indentation- and voltage-dependent fashion.

To complement the ATP release studies, we tested the converse by degrading ATP using the enzyme apyrase (*Palygin et al., 2015*, *Palygin et al., 2017*). Apyrase decreased innocuous and noxious mechanical responses in evoked behavior assays in vivo, reduced mechanically evoked action potential frequencies for all primary afferent fiber types (C, Aδ and Aβ-fibers) tested in skin-nerve recordings, and diminished inward currents in P2X2 expressing HEK cells during mechanical stimulation of keratinocytes. Collectively, these data suggest that graded ATP signaling is essential for the transmission of mechanically relevant information between keratinocytes and sensory neurons. While platelets, fibroblasts and neurons also release ATP and are present in the skin (*Fukami and Salganicoff, 1977*; *Grierson and Meldolesi, 1995*; *Lazarowski et al., 2003*; *Abbracchio et al., 2009*), it is unlikely that ATP stores from these cells are critical for mechanotransduction since in vivo apyrase treatment did not amplify the effects of specific optogenetic inhibition of keratinocytes. Together, these data indicate that during mechanical stimulation (1)

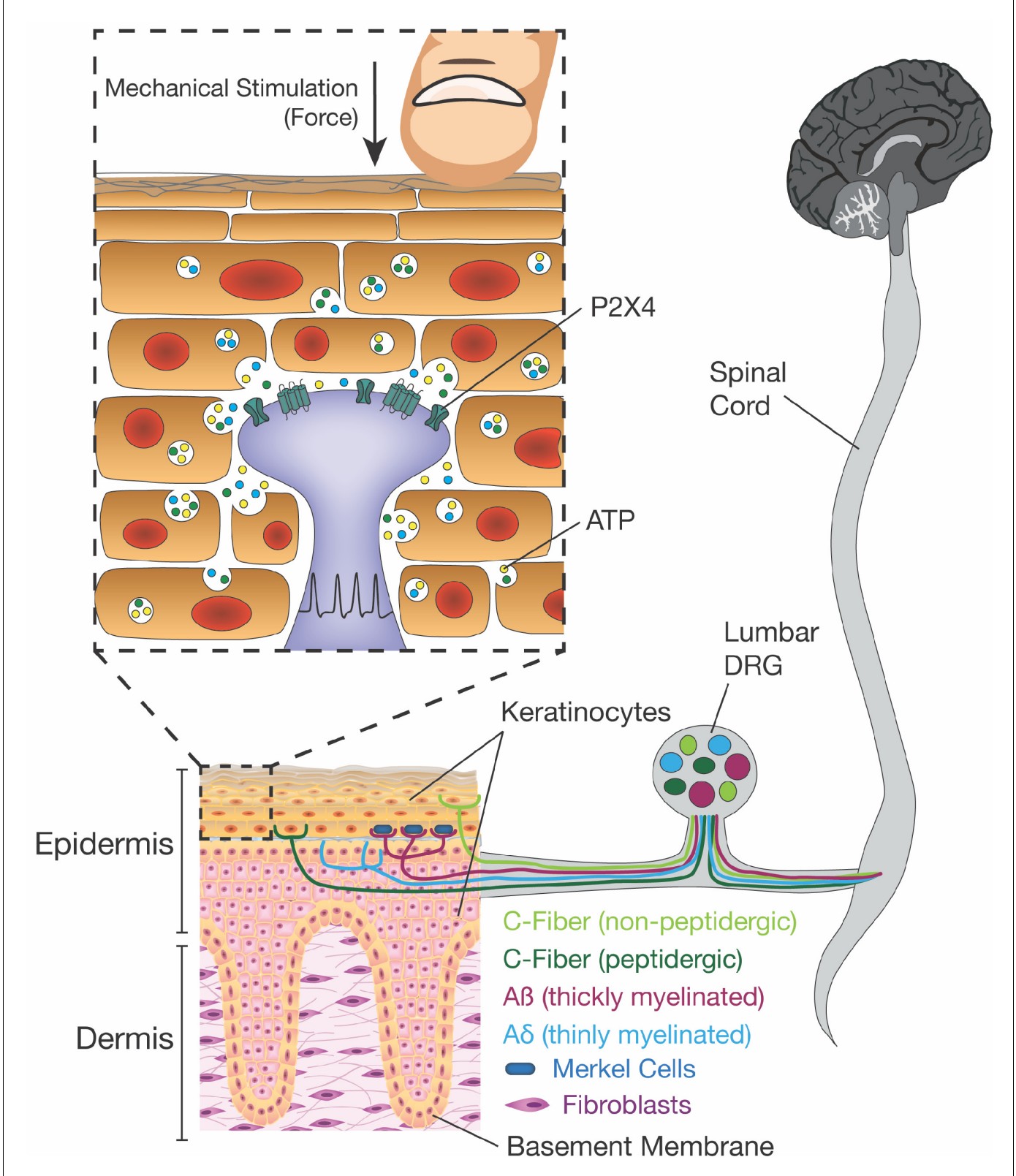

**Figure 9.** Schematic diagram depicting the proposed mechanism for ATP release induced by mechanical stimulation of keratinocytes and its interaction with P2X4 on sensory nerve endings. Touching of the skin, and therefore the mechanical stimulation of keratinocytes, elicits release of factors such as ATP, which in turn, acts on P2X4 and possibly other receptors on sensory neurons found within the epidermis, thereby causing action potential firing in the neurons and downstream effects leading to touch perception.

*Figure 9 continued on next page*

*Figure 9 continued*

DOI: https://doi.org/10.7554/eLife.31684.019

ATP is a key signaling molecule released from keratinocytes, and (2) keratinocytes are the predominant source of ATP that is released.

Additionally, our findings in the Archaerhodopsin cell sniff assay coupled with the data showing that keratinocytes depolarize in an indentation-dependent manner in response to mechanical stimulation, argue that the release of ATP from keratinocytes in response to mechanical stimuli is largely voltage-dependent, thereby, pointing towards a vesicular release mechanism for ATP. Other studies utilizing normal human epidermal keratinocytes suggest that that ATP can be released via a calcium-dependent vesicular mechanism and/or non-vesicular mechanism via connexin hemichannels (*Harden and Lazarowski, 1999*; *Lazarowski et al., 2003*; *Inoue et al., 2014*; *Barr et al., 2013*). Further studies are needed to identify the exact ATP release mechanism via mechanical stimulation of skin and primary mouse keratinocytes.

## Normal tactile sensation requires sensory neuron P2X4 expression

The keratinocyte-released ATP must be acting through a specific receptor or set of receptors on sensory nerve terminals in order to convey the innocuous and noxious touch signal(s) to the spinal cord. Our studies estimate the amount of ATP release to be in the micromolar range, which is an amount sufficient to activate most P2X receptors (*Jacobson et al., 2002*); however, the levels of ATP we measured are likely underestimations given that both keratinocytes and sensory neurons express ectonucleotidases (*Lazarowski et al., 2000*; *Zylka et al., 2008*; *Sowa et al., 2010a*; *Sowa et al., 2010b*). Additionally, it is possible that ATP is focally released in high concentration pocket 'domains' between the sensory neuron and keratinocyte cell membranes, and therefore, the ATP concentrations that occur in those localized signaling regions might be much higher than the generalized levels we measured in our assays. Although there are a number of P2X channels that have been shown to be expressed by sensory neurons (*Kobayashi et al., 2005*), we chose to investigate P2X4 because it is the most highly expressed P2X receptor on sensory neurons and because of its relatively equal expression on both light touch and nociceptive neurons (*Kobayashi et al., 2005*). Indeed, both pharmacological inhibition and genetic ablation of P2X4 in sensory neurons reduced tactile thresholds, blunted suprathreshold responses, and dampened responses to noxious pin prick. Accordingly, activation of P2X4 via a positive allosteric modulator decreased mechanical thresholds, and increased responses to suprathreshold stimuli, but had no effect in P2X4-deficient mice. Furthermore, the ablation of P2X4 in sensory neurons decreased mechanical responsiveness of primary afferent fibers in the ex vivo skin nerve preparation, as reflected by both significantly elevated thresholds and decreased afferent firing, especially at the higher intensity stimulus for all slowly adapting fiber types tested. These data indicate that the ATP released from keratinocytes is most likely signaling to P2X4 receptors on sensory neurons.

Peripheral inhibition of P2X2 and P2X3 receptors, which are also highly expressed on sensory neurons (*Kobayashi et al., 2005*) and have been shown to be involved in various pain states (*Novakovic et al., 1999*; *Cockayne et al., 2000*; *North, 2004*; *Bernier et al., 2017*), had no effect on baseline tactile thresholds or responses to a suprathreshold stimulus. Furthermore, in vivo degradation of ATP had no additional behavioral effect in sensory neuron-specific P2X4 knockout mice, therefore indicating that ATP signaling occurs mainly through P2X4 receptors on sensory neurons. It may be surprising that P2X4 was identified in our study as the key target of mechanically released ATP. However, purinergic signaling appears to be more multifaceted than would be expected by simply determining the probability of ATP binding via $EC_{50}$ values because (1) receptors can also exist in heteromeric confirmations and (2) P2 receptors have been shown to have complex response patterns, where rather than having distinct individual roles, different P2 receptors have been shown to work in concert through having both additive and inhibitory interactions (*Xing et al., 2016*).

## Keratinocyte-sensory neuron signaling serves as an potentiator of touch transduction

Sensory neurons are intrinsically capable of sensing mechanical stimuli via the activation of mechano-sensitive ion channels found on their terminals. Studies have identified Piezo2 and Transient Receptor Potential Ankyrin 1 (TRPA1) as key mechanosensitive ion channels found on sensory neurons (*Coste et al., 2010*; *Kwan et al., 2009*); *Vilceanu and Stucky, 2010*; *Woo et al., 2015*). ATP is likely a potentiator, rather than an initiator, of action potential firing in response to mechanical stimulation of the skin because we observed blunting of behavioral and afferent mechanical responses and decreased sensitivity thresholds in the various assays instead of finding a complete lack of responsiveness to force. Furthermore, in addition to ATP, keratinocytes can release a variety of chemical factors such as CGRPβ, β-endorphins, endothelin-1, neurotrophins, and cytokines (*Lumpkin and Caterina, 2007*; *Hou et al., 2011*; *Shi et al., 2013*), all of which can activate receptors on sensory nerve terminals. However, our data utilizing Arch-inhibition in combination with apyrase shows that ATP most likely is the major molecule released from keratinocytes upon innocuous and noxious touch at baseline in non-injured skin. However, after skin injury or disease, it is possible that one or more of these factors potentiates signals after injury in addition to ATP. If true, this injury-induced augmentation of keratinocyte communication could provide sensory neurons with numerous opportunities for initiation and amplification of signaling mechanisms that underlie pain, itch or dysesthesia in the setting of disease.

## Keratinocytes communication is not fiber type specific

Through the use of a glabrous skin-tibial nerve preparation, we show that either ATP hydrolysis or genetic knockdown of P2X4 in sensory neurons diminishes mechanically evoked activity in all primary afferent fiber types tested. It should be noted that the dampening of the afferent firing rate was much more prominent in the genetic P2X4 mutant model than in the experiment where apyrase was applied to the receptive fields via bath exposure. Moreover, effects on mechanical thresholds of fibers were observed in the genetic P2X4 mutants but not in the apyrase experiments. We believe that the lesser effect in the apyrase experiment is most likely due to the apyrase enzyme, which is diluted in aqueous buffer, not being able to penetrate and fully distribute within the tissue to the receptive terminals of sensory neurons. Nonetheless, in both the P2X4 mutant and apyrase teased fiber experiments, there were effects on all fiber types studied including slowly adapting Aβ, Aδ and C-fibers. These data indicate that keratinocytes are not only communicating with the more superficial non-peptidergic C-fibers (i.e. MrgD$^+$/IB4$^-$ binding C-fibers), but also signaling to the deeper-projecting peptidergic C-fibers, Aδ-fiber nociceptors and slowly adapting Aβ-fibers that mediate light touch, all of which are closer to the dermal-epidermal border in the skin (*Zylka et al., 2005*; *Basbaum et al., 2009*; *Abraira and Ginty, 2013*; *Le Pichon and Chesler, 2014*). These data are supported by previous studies, which also showed that optogenetic stimulation of K14-expressing cells activated slowly adapting Aδ and Aβ-fibers (*Baumbauer et al., 2015*) as well as C-fibers (*Baumbauer et al., 2015*; *Pang et al., 2015*). Consequently, these data together indicate that keratinocyte signaling is essential in potentiating signals of many fiber subtypes and that keratinocyte-sensory neuron communication is not a fiber-type-specific phenomenon.

Another interesting and novel finding is that all fiber types tested in P2X4 sensory neuron knockout animals also show elevated action potential thresholds, indicating decreased mechanical sensitivity at the terminal receptive field of single fibers. We have never before observed significant, potentially biologically relevant changes in fiber mechanical thresholds in any genetic mutant or injury study. The most parsimonious reason for this novel difference is that instead of using von Frey filaments as we have always done in the past, we utilized a new custom-designed feedback-controlled mechanical stimulator probe to exert a continuous force ramp from 0 to 100 mN perpendicularly to each fiber's skin receptive field. The surface area of the new stimulator probe utilized to exert this force is flat, circular and larger (0.8 mm diameter; approximately 2–4 times larger) than the tip of a typical von Frey filament which is small and pointed (4 mN: 0.19 mm diameter; 6 mN: 0.25 mm diameter). Thus, the new probe used for the ramp may stimulate the receptive field more evenly and consistently than the von Frey filament tip which may have sharper edges that deliver more punctate stimuli to the receptive field and activate the fiber at lower thresholds. This idea is generally supported by previous evidence that cutaneous sensory

terminals are finely tuned to detect and encode the edges of objects touched (*Wheat and Goodwin, 2001*).

## Foundation for future studies on tissue injury or disease

This is the first study that establishes a clear role for keratinocyte-initiated purinergic signaling in mechanotransduction at the cellular, tissue and behavioral levels. It has been shown that diseases such as complex regional pain syndrome and post-herpetic neuralgia are accompanied by increased epidermal ATP release which can lead to excessive activation of P2X receptors on sensory neurons (*Zhao et al., 2008*). Many other skin disorders, such as dermatitis and psoriasis, are characterized by altered keratinocyte function and signaling, and also share cutaneous pain as a common debilitating symptom that leads to severely decreased quality of life in affected patients (*Man, 2011*). If mechanical allodynia and hyperalgesia can effectively be treated at the site of pain (i.e. the skin, via interfering with keratinocyte-sensory neuron communication), it would allow for easy, non-invasive treatment options that avoid the central nervous system-mediated side effects of most current pain treatments, including opioid analgesic drugs. Our current study encourages further exploration into ATP and P2X4 as valuable targets for novel topical analgesics and antipruritics.

## Materials and methods

**Key resources table**

| Reagent type (species) or resource | Designation | Source or reference | Identifiers | Additional information |
|---|---|---|---|---|
| Strain, strain background (C57BL/6J) | C57BL/6J | The Jackson Laboratory | Jackson Stock #: 000664 RRID:IMSR_JAX:000664 | |
| Strain, strain background (K14Cre) | K14Cre | The Jackson Laboratory | B6N.Cg-Tg(KRT14-cre)1Amc (Jackson stock #: 018964) RRID:IMSR_JAX:018964 | |
| Strain, strain background (Archaerhodopsin3 fl/fl) | Arch | The Jackson Laboratory | Ai35D (B6;129S-Gt(ROSA) 26Sor$^{tm35.1(CAG-aop3/GFP)Hze}$/J (Jackson stock #: 012735) RRID:IMSR_JAX:012735 | |
| Strain, strain background (Channelrhodopsin2 fl/fl) | ChR | The Jackson Laboratory | Ai32 (B6;129S-Gt(ROSA) 26Sor$^{tm32(CAG-COP4+H134R/EYFP)Hze}$/J (Jackson stock #: 007909) RRID:IMSR_JAX:007909 | |
| Strain, strain background (TdTomato fl/fl) | tdTomato | The Jackson Laboratory | Ai14; B6.Cg-Gt(ROSA) 26Sor$^{tm14(CAG-TDTomato)Hze}$/J (Jackson stock #: 007914) RRID:IMSR_JAX:007914 | |
| Strain, strain background (AdvillinCre) | Advillin Cre | doi: 10.1073/pnas.1014411108; PMCID: PMC3044401 | | |
| Strain, strain background (P2X4 fl/fl) | P2X4 | doi: 10.1161/CIRCHEARTFAILURE. 113.001023; PMCID: PMC4289151 | | |
| Cell line (293 [HEK-293] cell line) | HEK-293 cells | ATCC® | RRID:CVCL_0045 Cat# ATCC® CRL-1573™ | |
| Transfected construct (GFP-tagged P2RX2 plasmid) | GFP-P2X2 HEK-293 cell | Origene | RG216207 | |
| Antibody (Rabbit polyclonal anti-Keratin 14) | K14 | Biologend | RRID:AB_2616896 Cat# 905304 AB_2616894 | 1:500 dilution |
| Antibody (Donkey anti-rabbit AlexaFluor 594) | | Invitrogen | RRID:AB_2556547 Cat# R37119 | 2 drops/1mL |
| Commercial assay or kit (Purelink RNA Microscale Kit) | | ThermoFisher Scientific | Cat#12183016 | |

*Continued on next page*

*Continued*

| Reagent type (species) or resource | Designation | Source or reference | Identifiers | Additional information |
|---|---|---|---|---|
| Commercial assay or kit (Superscript Variable Input Linear Output (VILO) cDNA Synthesis Kit) | | ThermoFisher Scientific | Cat#11754050 | |
| Commercial assay or kit (TaqMan primer and probes mouse P2X4) | | LifeTechnologies | Assay ID: Mm00501787_m1 | |
| Commercial assay or kit (TaqMan primer and probes mouse GAPDH) | | LifeTechnologies | Assay ID: Mm99999915_g1 | |
| Commercial assay or kit (Sarissa Probe® ATP / ATP) | | Sarissa Biomedical Limited; doi: 10.3791/53059; PMCID: PMC4684070 | Cat# SBS-ATP-05-50 | |
| Commercial assay or kit (Sarissa Probe® Null / NUL) | | Sarissa Biomedical Limited; doi: 10.3791/53059; PMCID: PMC4684070 | Cat# SBS-NUL-05-50 | |
| Chemical compound, drug (apyrase) | apyrase | Sigma-Aldrich; doi:10.1371/journal.pbio.1001747 | CAS. No. 9000-95-7 Cat#A6237 | |
| Chemical compound, drug (5-BDBD (5-(3-Bromophenyl)-1,3-dihydro-2H-benzofuro[3,2-e]-1,4-diazepin-2-one)) | 5-BDBD | Tocris Bioscience | CAS 768404-03-1 Cat# 3579 | |
| Chemical compound, drug (ivermectin) | IVM or ivermectin | Tocris Bioscience | CAS 70288-86-7 Cat#1260 | |
| chemical compound, drug (dimethyl sulfoxide) | DMSO | Sigma-Aldrich | Cat # 8418-100mL | |
| Chemical compound, drug (NF 110) | NF 110 | Tocris Bioscience | CAS 111150-22-2 Cat#2548 | |
| Software, algorithm (Pulse) | | HEKA Electronics | http://www.heka.com/downloads/downloads_main.html#down_patchmaster | |
| Software, algorithm (FitMaster) | | HEKA Electronics | http://www.heka.com/downloads/downloads_main.html#down_fitmaster | |
| Software, algorithm (Origin 5.0) | | Origin Lab | http://originlab.com | |
| Software, algorithm (Graphpad Prism 7) | | Graphpad | https://graphpad.com/scientific-software/prism/ | |
| Software, algorithm (ANY-maze) | | ANY-maze | http://www.anymaze.co.uk/index.htm | |
| Software, algorithm (LabChart) | | ADInstruments | https://www.adinstruments.com/products/labchart | |
| Software, algorithm (DY2000 Multi-channel Potentiostat) | | Digilvy | http://www.digi-ivy.com/dy2000.html | |
| Other (490 nm LED) | 490 nm | Thorlabs inc | Cat# M490L2 | |
| Other (590 nm LED) | 590 nm | Thorlabs inc | Cat# M590L2 | |
| Other (Compact T-Cube LED driver) | | Thorlabs inc | CAT# LEDD1B | |
| Other (473 nm Laser) | 473 nm | Laserglow | Cat# R471005GX | |
| Other (589 nm Laser) | 589 nm | Laserglow | Cat# R581005GX | |
| Other (595 nm LED strip) | 596 nm LED floor | Environmental Lights | Cat# amber3528-120-10-reel | |
| Other (460 nm LED strip) | 460 nm LED floor | Environmental Lights | Cat# blue3528-120-10-reel | |
| Other (BAT-12 Microprobe Thermometer) | | Physitemp | Cat# BAT-12 | |

*Continued on next page*

*Continued*

| Reagent type (species) or resource | Designation | Source or reference | Identifiers | Additional information |
|---|---|---|---|---|
| Other (DY2023 multi-channel potentiostat) | | Digilvy | Cat# DY2023 | |
| Other (EPC9 single patch amplifier) | | HEKA Electronics | Cat# EPC9 | |
| Other (ENV 800 SC amplifier module) | | Piezosystemjena | Cat# E-280-100 | |
| Other (PA 25114 SGpreloaded stack type actuator) | | Piezosystemjena | Cat# P-153-01 | |
| Other (Superfrost Plus Gold Slides) | | Electron Microscopy Sciences | Cat #: 71864-01 | |

## Animals

Adult male C57BL/6J mice from Jackson Laboratories (Jackson stock number 000664; Bar Harbor, Maine) of at least 8 weeks of age were used for pharmacological studies (apyrase, 5-BDBD, ivermectin with 5-BDBD, and NF 110), ATP biosensor, and skin nerve experiments. For all other studies male and female mice aged 7–16 weeks were used. Male and female mice were analyzed separately, however, since no sex differences were observed, the data for all studies show combined results of both sexes.

*Keratin14* (K14) is expressed in all keratinocytes as early as E9.5 (*Byrne et al., 1994*; *Dassule et al., 2000*; *Wang et al., 1997*). To create the mouse line that selectively expresses GFP-tagged Archaerhodopsin-3 in K14-positive cells, Ai35D (B6;129S-Gt(ROSA)26Sor$^{tm35.1(CAG-aop3/GFP)Hze}$/J) Archaerhodopsin (Jackson stock number 012735) and B6N.Cg-Tg(*KRT14*-cre)1Amc (Jackson stock number 018964) lines were mated. Offspring were genotyped as either *K14Cre$^+$ Arch/Arch* (Arch-K14Cre$^+$) or *K14Cre$^-$ Arch/Arch* (Arch-K14Cre$^-$). To create a complementary line that selectively expresses eYFP-tagged Channelrhodopsin-2 in K14-positive cells, Ai32 (B6;129S-Gt(ROSA)26Sor$^{tm32(CAG-COP4+H134R/EYFP)Hze}$/J) enhanced Channelrhodopsin 2 (Jackson stock number 012569) and B6N.Cg-Tg(*KRT14*-cre)1Amc (Jackson stock number 018964) lines were mated. Offspring were genotyped as either *K14Cre$^+$ChR/ChR* (ChR-K14Cre$^+$) or *K14Cre$^-$ ChR/ChR* (ChR-K14Cre$^-$). To create mice that selectively express tdTomato in Keratin14-expressing cells were created by Ai14; B6.Cg-Gt(ROSA)26Sor$^{tm14(CAG-TDTomato)Hze}$/J (Jackson stock number 007914) with B6N.Cg-Tg(*KRT14*-cre)1Amc (Jackson stock number 018964). Conditional *P2rx4* knockout animals were generated by mating Wang *Advillin Cre* mice (previously described in *da Silva et al., 2011*) with *P2rx4* animals, which were generously provided to us by Dr. Bruce Liang (*Yang et al., 2014*). Offspring were genotyped as either *AdvillinCre$^+$P2X4$^{fl/fl}$* (P2X4-AdvCre$^+$) and *AdvillinCre$^-$P2X4$^{fl/fl}$* (P2X4-AdvCre$^-$). As a note, *Advillin* has also been found to be expressed in Merkel cells of the glabrous epidermis, and has been uses as to create knockout models to study Merkel cells (*Ranade et al., 2014*). All genotypes were confirmed by PCR.

Animals were housed in a climate-controlled room with a 14:10 light:dark cycle, on Sani-Chips an aspen wood chip bedding (P.J. Murphy Forest and Products, New Jersey) with Enviro-dri nesting material (Shepherd Specialty Papers, Michigan) and *ad libitum* access to food and water. All animals were group housed. Animal procedures adhered to the NIH Guide for the Care and Use of Laboratory animals, and were performed in accordance with the Institutional Animal Care and Use Committee at the Medical College of Wisconsin (approval #: 0383). Animals were randomly assigned different treatment groups.

## Immunohistochemistry

Glabrous skin of the mouse hindpaw was dissected and tissue was fixed in 4% paraformaldehyde. After 30% sucrose cryoprotection, skin was processed into 6 µm sections using a ThermoFisher Scientific Microm HM550. Sections were blocked in 10% normal donkey serum, 0.3% Triton X-100 in PBS, then incubated overnight at 4°C in rabbit anti-keratin 14 primary antibody (1:500; AB_2616894; Biolegend, San Diego, CA). After rinsing, sections were incubated in donkey anti-rabbit AlexaFluor 594 (2 drops/1 mL; R37119; Invitrogen) for 30 min. Immunofluorescent images were taken on a

Nikon Eclipse E800 confocal microscope equipped Nikon EZ-C1 software (Nikon Instruments, Melville, NY). The experimenter was blinded to genotype throughout the staining and imaging procedure and at least 3 animals of each genotype were used. Images were assessed in ImageJ (National Institutes of Health, Bethesda, MD) and Adobe Illustrator software (Adobe Systems Inc., San Jose, CA).

## Primary keratinocyte cell culture

Glabrous skin was isolated from the hindpaw as described above and incubated in 10 mg/mL dispase (Gibco, ThermoFisher Scientific, Waltham, MA) for 45 min at room temperature. Epidermal sheets were peeled from the dermis, then incubated in 50% EDTA (Sigma-Aldrich) in Hanks' Balanced Salt Solution without calcium chloride, magnesium chloride and magnesium sulfate (Gibco) for 27 min at room temperature. Sheets were exposed to 15% heat inactivated fetal bovine serum (ThermoFisher Scientific, Carlsbad, CA) then rubbed against the base of a petri dish to separate the keratinocytes from the epidermal sheets. The mixture was then centrifuged, the supernatant removed and the pellet was then re-suspended and cells were grown in Epilife media (Gibco) supplemented with 1% human keratinocyte growth supplement (Gibco), 0.2% GibcoAmphotericin B (250 µg/mL of Amphotericin B and 205 µg/mL sodium deoxycholate, Gibco) and 0.25% penicillin-streptomycin (Gibco) on laminin coated coverslips. Plates were incubated and grown in 37°C and 5% $CO_2$ conditions. Growth media was exchanged every 2 days. Keratinocytes were used 3 days after plating. For primary keratinocyte cell cultures, a mixture of male and female mice were utilized, although no significant sex differences were observed and therefore, the data for all studies show combined results from both sexes.

## Whole cell patch clamp

DRG neurons after overnight culture and keratinocytes 3 days after plating were viewed on a Nikon Eclipse TE200 inverted microscope. Cells were continuously superfused with extracellular normal HEPES solution containing (in mM): 140 NaCl, KCl, 2 $CaCl_2$, 1 $MgCl_2$, 10 HEPES, and 10 glucose, pH 7.4 ± 0.05, and 310 ± 3 mOsm. GFP$^+$ HEK293 cells (holding voltage −40 mV), keratinocytes (holding voltage −50 mV) or DRG neurons (holding voltage −70 mV) were patched in voltage clamp mode with borosilicate glass pipettes (Sutter Instrument Company, Novato, CA) filled with intracellular normal HEPES solution containing (in mM): 135 KCl, 10 NaCl, 1 MgCl2, 1 EGTA, 0.2 NaGTP, 2.5 ATPNa$_2$, and 10 HEPES, pH 7.20 ± 0.05, and 290 ± 3 mOsm. Cell capacitance and series resistance were kept below 10 MΩ for all cell types. Mechanical stimulation of cells occurred at a rate of 106.25 µm/ms by a second borosilicate glass pipette that was driven by a piezo stack actuator (PA25, PiezoSystemjena, Jena, Germany). Cells were stimulated with increasing displacements of 1.7 µm/Volt over 200 ms. To avoid sensitization and/or desensitization of keratinocytes, 3 min were given between each displacement. Recordings were made in Pulse software via an EPC9 amplifier (HEKA Electronics, Holliston, MA). Cells were included in the study if the leak current stayed below 200 pA for at least three stimulations. Data were analyzed using FitMaster (HEKA Electronics).

## Patch clamp with light stimulation

Keratinocytes were used on day 3 days after plating. Keratinocytes from K14-Arch animals were patched in current clamp mode (held at 0 pA); resting membrane potentials were measured at baseline, during 1 min of 590 nm LED light (4 mW and 5 mW, Thorlabs Inc. Newton, NJ) exposure, and 1 min after 490 nm light cessation. Keratinocytes from K14-ChR animals were patched in voltage clamp mode (held at −50 mV) using the method described above. Cells were exposed to 490 nm LED light (490 nm; 0.2, 2, 3, and 20 mW, Thorlabs) in 30 second increments; peak and sustained currents were recorded.

## Behavior

In all assays, animals were randomly assigned to treatment groups; the experimenter was blinded to genotype and/or treatment. All assays were performed between 8 am and 2 pm, and mice were acclimated to their surroundings and the experimenter for at least 1 hr prior to testing. Adult male C57BL/6J mice from Jackson Laboratories of at least 8 weeks of age were used for pharmacological studies (apyrase, 5-BDBD, ivermectin with 5-BDBD, and NF 110), apyrase skin nerve and ATP

biosensor experiments. Approximately equal numbers of male and female mice were used for (K14-Arch, K14-ChR and P2X4-AdvCre) experiments, and because no significant differences were noted between the sexes, data from both sexes were combined.

## Mechanical sensitivity testing

Using the Up-Down method and a series of calibrated von Frey filaments ranging from 0.38-37 mN, mechanical thresholds of the glabrous hindpaw skin was assessed (*Chaplan et al., 1994*; *Dixon, 1980*). Mechanical responsiveness was also assessed using a suprathreshold 3.61 mN von Frey filament applied 10 times to the glabrous skin of the hindpaw and the number of stimulus-evoked paw withdrawals was recorded (*Weyer et al., 2016*). To assess noxious mechanical sensitivity, a spinal needle tip was applied to each of the plantar hindpaws; the number and type of responses to ten stimulations was recorded (*Hogan et al., 2004*; *Garrison et al., 2014*). Response categories include: normal/innocuous (simple withdrawal of the paw), noxious (elevating the paw for extended periods of time, flicking and licking of the paw), and null responses (no withdrawal).

## Optogenetic manipulations during behavioral testing

A 590 nm amber LED (17.5 mW; Thorlabs) was used to activate Arch$^+$ cells; a 490 nm blue LED (21.6 mW; Thorlabs) served as a control in Arch-based experiments. During mechanical sensitivity testing, LEDs were placed roughly 5–6 cm beneath the testing platform. The hindpaw was exposed to the LED for 1 min before and during the length of each mechanical test.

To determine the effects of keratinocyte activation on sensory-related behaviors, hindpaws were exposed to a 473 nm blue laser (10 Hz frequency, 24–28 mW; Laserglow, Canada) or control light from a 589 nm amber laser (10 Hz frequency, 15–20 mW; Laserglow) and the animal behavior was videotaped. Lasers were coupled to an optical fiber with a fiber coupler which was held 6–8 cm below the hindpaw. Behaviors were recorded during each 6 min optical stimulation period; videos were analyzed offline by an experimenter blinded to both genotype and treatment. Behaviors were timed and classified as follows: noxious (biting, licking hindpaw), front paw noxious (biting, licking front paw), and null response. Notably, animals also often attended to their front paw, even though that was not the target of the light stimulation; this occurred because as animals were attending to their hindpaw they often lifted the hindpaw and held it in both forepaws, thereby also exposing the forepaw to the light stimulus.

## Pharmacological treatment during behavioral testing

Apyrase (0.4 units Sigma-Aldrich, St Louis MO) or vehicle (PBS, Gibco) was injected into the plantar surface of the hindpaw 45 min prior to testing. Apyrase is an enzyme that catalyzes ATP into AMP and inorganic phosphate and its specificity has been shown previously (*Palygin et al., 2015*; *Palygin et al., 2017*). 5-BDBD (5-(3-Bromophenyl)-1,3-dihydro-2H-benzofuro[3,2-e]-1,4-diazepin-2-one, 10 mM, Tocris Bioscience, Avonmouth Bristol, United Kingdom) or vehicle (DMSO, Sigma-Aldrich) was similarly injected 1 hr prior to testing. Ivermectin (900 µM Tocris Bioscience) or vehicle (75% DMSO (Sigma-Aldrich), 25% PBS (Gibco) was injected 20 min prior to testing. Low 500 nM NF 110, high 5 mM NF 110 (Tocris Bioscience) or vehicle (PBS, Gibco) was injected into the plantar surface of the hindpaw 60 min prior to testing. Injection volume of 30 µL was used unless specified otherwise. The experimenter was blinded to treatment.

## Real-time optogenetic place preference assay

Animals were placed in a two-chamber Plexiglas box; chambers were divided by an opaque black Plexiglas sheet with a 5.1 × 5.1 cm opening at the bottom that acted as a passageway between the two chambers. The LED floor was constructed using 595 nm and 460 nm LED strip lights (595 nm: 278 µW, 460 nm: 75.2 µW; Environmental Lights, San Diego, CA). Each box was equipped with a fan on the right side and lights on each side of the box. Mice were acclimated to the box for 10 min and then the floor diodes were turned on for 30 min. Movement was recorded and analyzed by ANY-maze tracking software (ANY-maze, Wood Dale, IL). K14-ChR animal videos were subsequently analyzed by a blinded experimenter who recorded the amount of time each animal spent grooming in the individual chambers.

## Paw temperature measurements

To determine if significant heating of the paw occurred in response to either the LED or laser stimulation, an implantable thermocouple microprobe was inserted into the glabrous skin (Physitemp; Clifton, NJ). For this procedure, animals were anesthetized with 1.5% isoflurane; body temperature was measured throughout the procedure and maintained with a heating pad. Laser coupled fiber optics and LEDs were held at the appropriate distance from the hindpaw and temperature measurements were made over a 6 min time window.

## Teased fiber skin-nerve recordings

To assess changes in the presence or absence of apyrase of primary afferent firing, we utilized tibial skin-nerve preparations, as described (*Reeh, 1988*). We chose to use the tibial nerve because it innervates a majority of the glabrous skin of the mouse hindpaw, which was tested in all behavior assays. Animals were briefly anesthetized and then sacrificed via cervical dislocation. The leg of the animal was then shaved with commercial clippers, and the glabrous skin and tibial nerve was quickly removed and placed in a heated (32 +- 0.5°C), oxygenated bath consisting of (in mM): 123 NaCl, 3.5 KCl, 2.0 CaCl$_2$, 0.7 MgSO$_4$, 1.7 NaH$_2$PO$_4$, 5.5 glucose, 7.5 sucrose 9.5 sodium gluconate and 10 HEPES. The buffer pH was then adjusted to a pH of 7.45 +- 0.05. Either PBS or 40 units of apyrase were added to the bath where the skin was kept (experimenter was blinded to the treatment group). To keep the skin in place it was pinned down with insect needles, and the tibial nerve was placed in a chamber with a mirror plate. The nerve end was kept on the mirror plate surrounded by mineral oil while it was being teased into small fascicles. These small bundles were then placed on the recording electrode and a blunt glass probe was used to mechanically stimulate the preparation in order to find receptive fields of single afferent fibers. Fibers were characterized based on their shape and conduction velocities: C-fibers < 1.2 m/s; Aδ-fibers 1.2–10 m/s; and Aβ-fibers for conduction velocities over 10 m/s (*Koltzenburg et al., 1997*). Of note, in the glabrous skin-tibial nerve preparation, all Aδ fibers we encountered were slowly adapting, and the majority of Aβ fibers we encountered were slowly adapting. Thus, only slowly adapting afferents were included in this study. To determine action potential firing thresholds, von Frey filaments were utilized in apyrase experiments. However, action potential thresholds in P2X4-AdvCre experiments were determined using a continuous force ramp from 0 to 100 mN utilizing a new custom designed feedback-controlled mechanical stimulator. Once a fiber was identified a baseline recording of their firing activity was recorded for 2 min. Next, a feedback-controlled mechanical stimulator was placed over the receptive field to stimulate it with increasing forces. For apyrase studies, the receptive field was stimulated with 20, 40, 75, and 110 mN for 12 seconds for C-fibers and for Aδ-fibers and Aβ-fibers with 25, 40, 75, 110, and 150 mN for 12 seconds. For P2X4-AdvCre experiments a new custom designed closed-loop feedback-controlled mechanical stimulator was used, which consists of three motorized and linear stages (T-LSM200A, Zaber Technologies Inc., Vancouver, BC, Canada) configured as a Cartesian (x,y,z) gantry. Using an ultra low force transducer (F30, Harvard Apparatus, Holliston, MA) mounted to the vertical, z-axis of this gantry, mechanical stimulations (2, 5, 10, 20, 40, 100 and 150 mN for 10 seconds) of the receptive field were performed. To prevent sensitization and desensitization of the fiber, a 1 min interval was given between the different forces. Data was recorded via Labchart (ADInstruments; Colorado Springs, CO).

## Ex vivo measurements of ATP release from skin

### ATP biosensors

Purine biosensors (Sarissa Biomedical Limited, Conventry, England) were used in a dual simultaneous amperometric recording set up. A null probe lacking the enzymatic biolayer was used to control for non-specific recordings including artifacts from skin movement, temperature, and pH. The second probe was a platinum microelectrode ATP biosensor covered in an ultrathin biolayer containing two enzymes, glycerol kinase and glycerol-3-phosphate oxidase. The former enzyme converts extracellular ATP and glycerol into ADP and glycerol-3-phosphate; these products are subsequently converted to glycerone phosphate and H$_2$O$_2$ by the latter enzyme. H$_2$O$_2$ is ultimately detected via oxidation of the electrode. These ATP sensors respond rapidly (10–90% rise in <10 s) and in a linear manner to physiologically relevant ATP concentrations (*Llaudet et al., 2005*). During data collection, the biosensors were connected to a multi-channel potentiostat and recording system (Digi-Ivy, Inc., Austin,

TX). Microelectrodes were calibrated to known ATP concentrations before and after each experiment in physiological buffer consisting of (in mM): 123 NaCl, 3.5 KCl, 2.0 CaCl$_2$, 0.7 MgSO$_4$, 1.7 NaH$_2$PO$_4$, 5.5 glucose, 7.5 sucrose 9.5 sodium gluconate, 10 HEPES and 2% glycerol, pH 7.45 ± 0.05. To increase sensitivity, biosensors were cycled from −500 mV to +500 mV at a rate of 100 mV/s for 10 cycles (Zappia et al., 2016). The sensors were polarized to +600 mV relative to an Ag/AgCl potentiostat reference electrode that was placed onto the skin.

## Mechanical stimulation of isolated skin

Glabrous paw skin was excised from the hindpaw without muscle, tendons, or blood vessels attached. Skin was stabilized with an anchor grid on Sylgard-coated petri dishes (184 silicone elastomer base; Dow corning corporation, Midland, MI). The biosensor probe tips (50 µm diameter, 500 µm length) were bent so that the sensing portion could be laid flat against the inner surface of the excised glabrous skin. Skin was then mechanically stimulated using calibrated von Frey filaments ranging from 1.6 to 84.8 mN force. During each 10 second stimulation period, the appropriate filament was applied rapidly and repeatedly (15–20 applications). Sixty seconds passed between each stimulation period. Fresh oxygenated buffer was applied every 15 min. To quantify ATP release, null probe traces were first subtracted from ATP probe traces for a given stimulation period. Resulting values were extrapolated from calibration response ratios generated with known ATP concentrations. utilizing Origin 5.0, and the area under the curve for each trace and mechanical stimulation was determined to evaluate total ATP release.

## Cell lines

### HEK293 cell culture

HEK-293T cells were purchased and certified from ATCC (ATCC, Gaithersburg, MD). This cell line was used because it is a highly transfectable cell line that is commonly used with this type of assay. The cells were negative for mycoplasma. HEK293 cells, cells were grown in T-25 and T-75 flasks (VWR, Wayne, PA) in DMEM media containing 4.5 g/L D-Glucose, 4.5 g/L L-glutamine and 110 mg/L sodium pyruvate (Gibco), supplemented with 1% Penicillin-Streptomycin (10,000 U/mL, Gibco) and 10% Fetal Bovine Serum (Gibco) in 37°C and 5% CO$_2$ conditions. The media was exchanged every 2–3 days. Cells were split once they reached 85–90% confluency.

## HEK293 cell transfection

In order to develop a cell line that over expresses *P2rx2*, 1 million HEK293 (ATCC) cells were transfected with 500 ng of a plasmid expressing a C-terminal GFP-tagged *P2rx2* plasmid (RG216207 Origene, Rockville, MD) using the Lonza 4D nucleofector (Basel, Switzerland). P2X2 receptors were chosen due to their favorable ion channel kinetics (Coddou et al., 2011). Following transfection, cells were sorted on a FACSARIA cell sorter (San Jose, CA) to select for GFP-expressing cells 72 hr post transfection. To maintain the cells, once every other week, cells were sorted for GFP expression to establish a cell line that has stable GFP-tagged *P2rx2* overexpression. GFP$^-$ cells in the culture were used as internal controls for the naive cell sniff assay.

## Cell sniff assay

Primary mouse keratinocytes from K14-tdTomato Cre$^+$ animals were cultured for at least 3 days before being utilized in the cell sniff assay. HEK293 (ATCC) cells transfected with P2X2 GFP construct (Origene) in close proximity to the keratinocytes. The experimenter was blinded to which transfected HEK cell line was utilized in the experiment. Round coverslips containing mixed cultures were viewed on a Nikon Eclipse TE200 inverted microscope. Keratinocytes in close proximity (1–15 µm) to patched P2X2-GFP$^+$ and GFP$^-$ HEK293 were mechanically stimulated as described above. When optogenetic mouse lines were utilized, either a 590 nm (5 mW) or 490 nm (0.2–20 mW) LED was mounted on top of the microscope and turned on during recordings.

## Dorsal root ganglia cell culture

Mice were euthanized and sensory neurons were isolated from bilateral lumbar 1–6 ganglia. Isolated DRG were incubated with 1 mg/mL collagenase type IV (Sigma, St. Louis, MO) for 40 min at 37°C and 5% CO$_2$, followed by a 45 min incubation with 0.05% trypsin (Sigma). The DRG were then

mechanically dissociated and plated onto laminin-coated glass coverslips. Two hours after plating, neurons were fed with media containing Dulbecco's modified Eagle's medium/Ham's F12 medium, supplemented with 10% heat-inactivated horse serum, 2 mM L-glutamine, 1% glucose, 100 units/ml penicillin, and 100 µg/ml streptomycin. The media contained no exogenous growth factors.

## Quantitative real-time polymerase chain reaction (qRT-PCR)

RNA was obtained from DRG samples by manually homogenizing tissue in Trizol then extracting nucleic acids using the Purelink RNA Micro Scale kit (ThermoFisher Scientific). All RNA samples were reverse transcribed into cDNA using the Superscript Variable Input Linear Output (VILO) cDNA Synthesis Kit (ThermoFisher Scientific). qRT-PCR was performed on a Bio-Rad C1000 Touch CFX96 Real-Time-System Thermal Cycler (Bio-Rad Laboratories Inc., Hercules, CA) using the following TaqMan primers and probes (LifeTechnologies) for mouse P2X4 and mouse GAPDH.

## Statistical analysis

Paw withdrawal thresholds and suprathreshold stimulus responses were compared between two groups using non-parametrcic Mann-Whitney U-test, and between three groups using a Kruskal-Wallis test. For groups that had a two by two set up, a two-way ANOVA with *Tukey* post-hoc test was used. For datasets that had a two by two by two set up, a three-way ANOVA with *Tukey* post-hoc test was used. Skin nerve recordings were analyzed using a repeated measures two-way ANOVA with *Sidak* post-hoc test. Types of responses to laser stimulation were analyzed using Chi square test with Fisher's exact for groupsof two, and Dirichlet-multinomial regression using the FMM procedure in SAS 9.4 (SAS Insitute, Cary, NC) for multiple groups, such as after apyrase/PBS treatment. Gene expression was analyzed using a two-tailed parametric t-test. Rheobase values were analyzed using a Mann-Whitney U-test, and resting membrane potentials were analyzed using a two-tailed parametric t-test. Summarized data are reported as mean ± SEM except for apyrase and PBS von Frey thresholds for the different fiber types which are reported as median ± interquartile range in the text. For all behavior experiments 'n' corresponds to the number of animals. For patch clamp studies, skin nerve recordings or ATP measurements at least n = 3 animals were utilized in each group analyzed, and the n on the graph corresponds to the number of cells, fibers, or repetitions. All data analyses were performed using Prism 7 software (GraphPad, La Jolla, CA), with an alpha value of 0.05 set *a priori.* *p<0.05 **p<0.01, ***p<0.001, ****p<0.0001, n.s. denotes a non-significant comparison.

## Acknowledgements

The authors thank Dr. Katelyn Sadler and Dr. Katherine J Zappia who provided advice on data analysis, organization and direction of the manuscript as well as Melissa Stagg, who scored the grooming behavior of the place preference videos. Furthermore, the authors would like to thank Margaret Beatka for the help and creativity in creating the scientific illustration in *Figure 9* of this manuscript, Jonathon Kokott and Dr. Larry Fennigkoh for building the newly designed custom feedback-controlled mechanical stimulator utilized for the P2X4-AdvCre studies, Dr. Aniko Szabo and Daniel Eastwood for providing assistance and consultation with the statistical analysis, as well as Dr. Ken Allen for helping to interpret the videos of animal behavior in response to activation of Channelrhodopsin in K14-expressing cells. The research reported in this manuscript was supported by NIH grants NS040538 and NS070711 to CLS. The authors would also like to thank the 'Research and Education Initiative Fund,' a component of the Advancing a Healthier Wisconsin Endowment at the Medical College of Wisconsin for funding and for allowing us to use the conditioned place preference apparatus in the Medical College of Wisconsin Neuroscience Research Core. The authors also thank the Marvin Wagner Endowed Professorship for funding.

## Additional information

### Funding

| Funder | Grant reference number | Author |
|---|---|---|
| National Institute of Neurological Disorders and Stroke | NS040538 | Cheryl L Stucky |
| National Institute of Neurological Disorders and Stroke | NS070711 | Cheryl L Stucky |
| Medical College of Wisconsin | Advancing a Healthier Wisconsin Endowment | Cheryl L Stucky |
| Marvin Wagner Endowed Professorship | | Cheryl L Stucky |

The funders had no role in study design, data collection and interpretation, or the decision to submit the work for publication.

### Author contributions

Francie Moehring, Conceptualization, Resources, Data curation, Software, Formal analysis, Supervision, Validation, Investigation, Visualization, Methodology, Writing—original draft, Project administration, Writing—review and editing; Ashley M Cowie, Anthony D Menzel, Andy D Weyer, Conceptualization, Data curation, Formal analysis, Writing—review and editing; Michael Grzybowski, Resources, Supervision, Validation, Methodology; Thiago Arzua, Data curation, Formal analysis; Aron M Geurts, Resources, Supervision, Validation; Oleg Palygin, Conceptualization, Resources, Data curation, Software, Formal analysis, Supervision, Validation, Methodology; Cheryl L Stucky, Conceptualization, Resources, Data curation, Software, Formal analysis, Supervision, Funding acquisition, Validation, Investigation, Visualization, Methodology, Writing—original draft, Project administration, Writing—review and editing

### Author ORCIDs

Francie Moehring (iD) http://orcid.org/0000-0002-0071-5685
Oleg Palygin (iD) http://orcid.org/0000-0002-3680-5527
Cheryl L Stucky (iD) http://orcid.org/0000-0003-4966-6594

### Ethics

Animal experimentation: All of the animal procedures strictly adhered to the NIH Guide for the Care and Use of Laboratory animals, and were performed in accordance with the Institutional Animal Care and Use Committee at the Medical College of Wisconsin (approval #: 0383).

### Decision letter and Author response

Decision letter https://doi.org/10.7554/eLife.31684.023
Author response https://doi.org/10.7554/eLife.31684.024

## Additional files

### Supplementary files

• Transparent reporting form
DOI: https://doi.org/10.7554/eLife.31684.020

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
