## [Decision Letter]

Thank you for submitting your article "Keratinocytes mediate tactile sensation via ATP-P2X4 signaling" for consideration by *eLife*. Your article has been favorably evaluated by Richard Aldrich (Senior Editor) and three reviewers, one of whom is a member of our Board of Reviewing Editors. The following individual involved in review of your submission has agreed to reveal his identity: Derek Moliver (Reviewer #2).

The reviewers have discussed the reviews with one another and the Reviewing Editor has drafted this decision to help you prepare a revised submission.

Summary:

This study explores the role of keratinocytes in tactile sensation using multiple approaches including optogenetics, electrophysiology, biochemistry and behavioral assessments. Prior work using optogenetic approaches has shown that keratinocytes play a role in modulating sensory functions. However, identifying chemical messenger(s) and receptors that are involved in keratinocyte-mediated modulation of sensory function provide new insights into this area. The results support a modulatory role of keratinocytes in mechanosensory transduction and implicate ATP as a mediator of keratinocyte-to-sensory neuron signaling, with P2X4 receptors in sensory neurons conveying this response.

Essential revisions:

While the reviewers are overall enthusiastic about the work, they have reached a consensus about a number of major concerns. Whether the paper is suitable for publication in *eLife* will depend on the authors' responses to these concerns.

1) Keratinocytes are not excitable cells, and the optogenetic methods used to hyperpolarize or depolarize keratinocytes may not mimic physiological responses of keratinocytes to mechanical forces acting on the skin. Therefore, the use of optogenetic methods in non-excitable cells needs to be justified. Can tactile stimulation of keratinocytes elicit membrane depolarization in these non-excitable cells? Does release of ATP from keratinocytes depend on keratinocyte depolarization?

In addressing this issue, the authors should discuss supporting evidence indicating expression of voltage-gated Na and Ca channels in keratinocytes and their functional significance (e.g., citing the work of Frank Rice et al.), as well as their capacity for Ca-dependent vesicular and non-vesicular release, including mechanically induced release. This work has been ongoing for several years in the purinergic field (e.g., Ken Harden Lab). The reviewers agree that a combination of additional experiments and proper review of the literature are needed to address this issue.

2) Figure 4 shows a critical series of experiments that test the role of endogenously released ATP from keratinocytes and P2X receptors on primary afferents in transduction. In addition to apyrase, pharmacological experiments with P2X receptor antagonists (e.g. blockers for P2X4 and P2X3 receptors) should be tested to confirm P2X receptor subtypes that are involved. P2X4-/- mice should also be used in these experiments. Also, regarding the data shown in Figure 4, average firing frequencies plotted in the graphs (C, E, G) show modest changes following the apyrase treatment, whereas the representative traces on the right (D, F, H) show examples that are quite dramatically different from controls. The authors should show more representative traces for these apyrase experiments.

3) Control experiments need to be done to determine specificity of the agents (apyrase, 5-BDBD, IVM) used in behavioral studies. The authors should determine whether high concentrations of these drugs may directly affect membrane excitability of sensory neurons. Additional controls for specificity of apyrase in particular would be a welcome addition to this study. It would also be helpful to provide some background on other efforts to use apyrase to block ATP signaling in the literature.

4) Interpretation of the behavioral results in Figure 2 is perplexing, as wiping and biting are widely considered nocifensive behaviors, but conditioned place aversion was not induced by laser stimulation. These findings also appear at odds with recent work from Caterina and colleagues (Pang et al.), which needs to be discussed. Given this, and the extensive expression of P2X and excitatory P2Y receptors on cutaneous nociceptors, it is therefore difficult to rule out a nociceptive basis for the behavioral results of Figure 2. In line with the stimulation of C-fibers reported here and the cited results of Beaudry et al., it is worth mentioning that Baumbauer et al. (cited elsewhere in the text) demonstrated activation of physiologically-identified nociceptors as well as other fiber types in response to optogenetic stimulation of keratinocytes. Can the authors discuss these discrepancies?

Other points to address:

1) How do the authors explain the long latency (avg 46 seconds) to elicit behavioral responses following keratinocyte activation? Is keratinocyte-to-sensory neuron transmission a sufficient mediator of sensory neuron activation or is it permissive? Does the long latency for behavioral responses following light stimulation support the latter? Please discuss.

2) Related to the above, the optogenic inhibition experiments shown in Figure 1 used light stimuli that were applied a minute before and during the mechanical stimulus. Why one minute before? Does inhibition only at the time of mechanical stimulation attenuate behavioral responses? And what is the effect of light stimulation in Arch-K14 mice on the physiological response properties, as measured in the apyrase experiments?

3) The absolute ATP concentrations are likely to be rough approximations and likely underestimates, given that both keratinocytes and sensory neurons are coated with ectonucleotidases. Please discuss in terms of the measured concentrations, compared to reported EC50 values for P2X receptors.

4) Related to major point 4 above, if ATP is indeed released from keratinocytes in response to innocuous tactile stimuli, then according to the authors' model a gentle touch would be expected to activate nociceptors and evoke pain because nociceptive afferents express P2X receptors and can be excited by ATP. This is obviously not the case under normal conditions. Please discuss.

5) The authors cite a paper showing that P2X4 receptor mRNA is expressed in most of sensory neurons. However, several previous studies showed that functional P2X receptors in DRG neurons are mainly P2X3, P2X2+3 receptors as determined by electrophysiological approaches. These P2X receptors are often co-localized with TRPV1 receptors in nociceptors. ATP-evoked currents mediated by P2X4 receptors have not been emphasized in those prior studies. Please discuss these points.

---

## [Author Response]

Essential revisions:

While the reviewers are overall enthusiastic about the work, they have reached a consensus about a number of major concerns. Whether the paper is suitable for publication in eLife will depend on the authors' responses to these concerns.1) Keratinocytes are not excitable cells, and the optogenetic methods used to hyperpolarize or depolarize keratinocytes may not mimic physiological responses of keratinocytes to mechanical forces acting on the skin. Therefore, the use of optogenetic methods in non-excitable cells needs to be justified. Can tactile stimulation of keratinocytes elicit membrane depolarization in these non-excitable cells? Does release of ATP from keratinocytes depend on keratinocyte depolarization?In addressing this issue, the authors should discuss supporting evidence indicating expression of voltage-gated Na and Ca channels in keratinocytes and their functional significance (e.g., citing the work of Frank Rice et al.), as well as their capacity for Ca-dependent vesicular and non-vesicular release, including mechanically induced release. This work has been ongoing for several years in the purinergic field (e.g., Ken Harden Lab). The reviewers agree that a combination of additional experiments and proper review of the literature are needed to address this issue.

We have added several pieces of key additional data to address this point.

a) “Can tactile stimulation of keratinocytes elicit membrane depolarization in these non-excitable cells?”

We thank the reviewers for this comment. To address this question, we performed patch clamp recordings from adult mouse keratinocytes in current clamp mode, and stimulated those with increasing focal mechanical force (new Figure 1). This data demonstrates that keratinocytes do depolarize in a graded manner in response to increasing focal indentation.

b) “…the use of optogenetic methods in non-excitable cells needs to be justified.”

We further conducted patch clamp recordings (in current clamp mode) from Arch-K14Cre^+^ keratinocytes with mechanical stimuli in the presence and absence of 590 nm light to activate Archaerhodopsin. The 590 nm light hyperpolarized keratinocytes at baseline and at every focal indentation from 0.84 to 5.88 µm (new Figure 1). We believe that this data justifies the use of optogenetics to modulate the membrane potential of keratinocytes.

c) “Does release of ATP from keratinocytes depend on keratinocyte depolarization?”

With this question, the reviewers raised a very interesting point. To answer this question, we measured ATP release in P2X2-transfected HEK293 cells that were co-cultured with ChR-K14Cre^+^ keratinocytes in the cell sniff assay. We exposed the co-cultured cells to either 590 nm light (Arch activation) or 490 nm light (ChR activation) at different power intensities. The negative control 590 nm did not elicit a current in P2X2-GFP^+^ HEK-293 cells, which reflects the absence of ATP release from keratinocytes under resting conditions (new Figure 3). In contrast, all 490 nm light intensities tested elicited inward current in P2X2-GFP^+^ HEK-293, reflecting ATP release from keratinocytes in response to depolarization. Furthermore, the current magnitude P2X2-GFP^+^ HEK-293 cells was light-intensity dependent such that more intense light elicited a greater current. These data indicate that ATP release from keratinocytes is elicited by depolarization in a stimulus-dependent manner. In addition, we also measured ATP a cell sniff assay with Arch-expressing keratinocytes in response to mechanical stimulation with the 590 nm light on or off. When the 590 nm light was on, and the keratinocytes hyperpolarized, we observed a significant decrease in ATP release suggesting that ATP release is largely voltage-dependent.

Overall, this data shows that ATP-release from keratinocytes is elicited largely by voltage-dependent events.

d) Discuss supporting evidence that keratinocytes express ion channels that could mediate depolarization and indicate potential mechanisms underlying the ATP release.

We have added additional discussion to cover these points and added additional references for Frank Rice and Kendall Harden.

“Native keratinocytes depolarize in response to mechanical stimuli

It may be surprising that keratinocyte function in vivo can be modulated by optogenetic manipulation of the membrane because these cells do not fire action potentials. […] These prior reports suggest that keratinocytes possess the functional ion channels required for producing rapid changes in membrane excitability, and our new data reveal that naïve adult mouse keratinocytes depolarize in response to mechanical stimuli.

“Additionally, our findings in the Archaerhodopsin cell sniff assay coupled with the data showing that keratinocytes depolarize in an indentation-dependent manner in response to mechanical stimulation, argue that the release of ATP from keratinocytes in response to mechanical stimuli is largely voltage-dependent, thereby, pointing towards a vesicular release mechanism for ATP. […] Further studies are needed to identify the exact ATP release mechanism via mechanical stimulation of skin and primary mouse keratinocytes.”

2) Figure 4 shows a critical series of experiments that test the role of endogenously released ATP from keratinocytes and P2X receptors on primary afferents in transduction. In addition to apyrase, pharmacological experiments with P2X receptor antagonists (e.g. blockers for P2X4 and P2X3 receptors) should be tested to confirm P2X receptor subtypes that are involved. P2X4-/- mice should also be used in these experiments. Also, regarding the data shown in Figure 4, average firing frequencies plotted in the graphs (C, E, G) show modest changes following the apyrase treatment, whereas the representative traces on the right (D, F, H) show examples that are quite dramatically different from controls. The authors should show more representative traces for these apyrase experiments.

We have added a new figure, Figure 8, and 2 panels to Figure 6 to address this point.

a) “Pharmacological experiments with P2X receptor antagonists (e.g. blockers for P2X4 and P2X3 receptors) should be tested to confirm P2X receptor subtypes that are involved.”

The reviewers raise an interesting point. To address it, we tested the effects of P2X2 and P2X3 by injecting the antagonist NF 110 (intraplantar) which inhibits P2X3 at a low concentrations (500 nM) and inhibits both P2X2 and P2X3 at a high concentrations (5 mM) in behavioral assays that measured mechanical thresholds and responses to suprathreshold force. Neither concentration had any effect on mechanical thresholds or responses to suprathreshold stimuli after 60 minutes (new Figure 6). Animals were also tested at both 30 and 120min after injection, and neither time point showed a significant difference in the mechanical assays used; 30min (Up-Down: p*=*0.9066 Kruskal-Wallis test; suprathreshold: p*=*0.6857 Kruskal-Wallis test;A and B on left) or 120min after injection (Up-Down: p*=*0.5808 Kruskal-Wallis test; suprathreshold p*=*0.4667 Kruskal-Wallis test, C and D on left). Data is not shown in the manuscript for these time points but included as Author response image 1. However, if the Editor or reviewers would like, we can include this data in a figure supplement. These data indicate that P2X4 (and not P2X3 or P2X2) receptors likely mediate the ATP-initiated responses in sensory neurons in non-injured tissue.

b) “P2X4-/- mice should also be used in these experiments.”

We thank the reviewers for this suggestion. In response, we performed skin nerve experiments in sensory neuron P2X4-deficient mice (new Figure 8). Exciting data shows that the knockdown of P2X4 in sensory neurons results in decreased mechanical responsiveness as reflected by both elevated thresholds (Figure 8) and decreased firing, particularly at higher-intensity stimuli (40 mN and above; Figure 8). All fiber types tested including C-fibers, SA-Aδ-fibers and SA-Aβ-fibers were affected. This data confirms the skin nerve results with apyrase where all fiber types were also affected by ATP hydrolysis in the skin.

b) Show more representative traces for the apyrase skin nerve recording data.

Thank you for this recommendation. We now provide recordings that closely represent the firing differences in the summary graphs for apyrase (Figure 4).

3) Control experiments need to be done to determine specificity of the agents (apyrase, 5-BDBD, IVM) used in behavioral studies. The authors should determine whether high concentrations of these drugs may directly affect membrane excitability of sensory neurons. Additional controls for specificity of apyrase in particular would be a welcome addition to this study. It would also be helpful to provide some background on other efforts to use apyrase to block ATP signaling in the literature.

We have added 2 new supplemental figures (Figure 4—figure supplement 1 and Figure 6—figure supplement 1) and one additional panel with new data to Figure 3 panel D.

a) “…determine whether high concentrations of these drugs may directly affect membrane excitability of sensory neurons.:”

We tested membrane excitability (rheobase) and resting membrane potential in patch clamp recordings of lumbar sensory neurons from naïve mice. High concentrations of apyrase (20 units) had no effect on either the rheobase or the resting membrane potential. These data indicate that apyrase does not likely affect the action potential threshold or alter the excitability of the sensory neuron membrane (Figure 4—figure supplement 1).

Furthermore, we tested the effects of high concentrations of 5-BDBD (20 mM) and ivermectin (9 mM) in patch clamp recordings. Neither compound had any effect on either action potential threshold or resting membrane potential (Figure 6—figure supplement 1). This suggests that high concentrations of these compounds do not likely alter the excitability of the sensory neuron membrane, and that the behavioral effects observed are most likely due to peripheral P2X4 inhibition (5-BDBD) and/or P2X4 potentiation (ivermectin).

b) “Additional controls for specificity of apyrase…”

We agree with the reviewers. To address this point, keratinocytes from naïve adult wild type mice were co-cultured with P2X2-transfected HEK-293 cells, and were mechanically stimulated in the presence of apyrase or vehicle in the cell sniff assay. Apyrase nearly abolished the inward currents in P2X2-GFP^+^ HEK cells in response to mechanical stimulation of keratinocytes (new Figure 4), thereby suggesting that apyrase in fact degraded the ATP released upon mechanical stimulation from keratinocytes.

Furthermore, we now include citations regarding the specificity of apyrase for ATP hydrolysis (Palygin et al. 2015; Palygin et al. 2017)in the Materials and methods as follows:

“Apyrase (0.4 units Sigma-Aldrich, St Louis MO) or vehicle (PBS, Gibco) was injected into the plantar surface of the hindpaw 45min prior to testing. Apyrase is an enzyme that catalyzes ATP into AMP and inorganic phosphate and its specificity has been shown previously (Palygin et al. 2015, 2017).”

4) Interpretation of the behavioral results in Figure 2 is perplexing, as wiping and biting are widely considered nocifensive behaviors, but conditioned place aversion was not induced by laser stimulation. These findings also appear at odds with recent work from Caterina and colleagues (Pang et al.), which needs to be discussed. Given this, and the extensive expression of P2X and excitatory P2Y receptors on cutaneous nociceptors, it is therefore difficult to rule out a nociceptive basis for the behavioral results of Figure 2. In line with the stimulation of C-fibers reported here and the cited results of Beaudry et al., it is worth mentioning that Baumbauer et al. (cited elsewhere in the text) demonstrated activation of physiologically-identified nociceptors as well as other fiber types in response to optogenetic stimulation of keratinocytes. Can the authors discuss these discrepancies?

We agree with the reviewers that these behavioral results are perplexing. We have included 9 new figure panels (Figure 1, updated Figure 2 (increased the number of animals), Figure 3, Figure 4, Figure 5, Figure 5—figure supplement 1, Figure 7) that specifically address these questions.

a) Why are nociceptive responses observed, but conditioned place aversion did not occur?

We are aware of the studies from the Albers and Caterina labs where keratinocytes were shown to have a role in nociceptor activation as well as A-fiber activation. We are also aware of the discrepancies in our data where in Figure 2 and Figure 2—figure supplement 1 we observe apparent nociceptive responses to K14-ChR activation, but alternatively do not observe place preference avoidance to the 460 nm floor in K14-ChR mice Figure 2. Instead of the nociceptive responses we see responses such as “grooming” and face wiping that more closely mimic animals that are experiencing paresthesia (Figure 2) (thanks to Dr. Ken Allen (Veterinarian at MCW) who helped us interpret the behaviors). We think the differences in these two behavioral assays are due to differences in the power intensity level used in either assay. In the assay measuring the evoked response to light (Figure 2) where apparent nociceptive responses were observed, the 473 nm power was significantly more intense at 25mW. In the place preference assay (Figure 2), where grooming and face wiping were observed without aversion, the 460 nm floor intensity was only 75.2 µW, a 333-fold intensity difference. New data on the differential effects of 490 nm light intensity are apparent in the ChR cell sniff assay (Figure 3). Therefore, a likely explanation is that the 460 nm floor with 75.2 µW power intensity is not strong enough to elicit enough ATP release to cause aversion to that side of the chamber. Another reason could be that in the assay measuring evoked responses, the 473 nm laser (25 mW power) was pulsed at 10 Hz (pulsing light stimulus), whereas the place preference floor LED lights were on throughout the entire 30 min while the animals were tested (i.e. a sustained light stimulus). This likely can also lead to differences in behaviors because when the light is pulsed, it is likely that ATP release is elicited in bursts, where peak intensities could be different; conversely, in with the sustained stimulus, the ATP release would most likely be sustained. Unfortunately, due to the lack of technical capabilities of the LED floor lights, we were unable to pulse the LED floor lights at the same 10Hz pulse as used in the evoked behavior assay.

b) To further address whether keratinocyte contribute to nociceptive responses, we performed additional experiments using a noxious mechanical assay where the plantar hindpaw is stimulated with a spinal needle tip. Indeed, 590 nm light activation of K14-Arch mice significantly reduced the noxious as well as innocuous responses and concomitantly increased null responses as compared to the 490 nm control light (Figure 1). Likewise, we now show that peripheral apyrase reduces the noxious and innocuous responses and increases null responses to needle probing (Figure 4). Combination of both 590 nm activation of keratinocytes and apyrase elicited no additional effect (Figure 5), as was observed for the likely innocuous mechanical assays of withdrawal threshold and suprathreshold response. In order to determine whether the mechanically-released ATP from keratinocytes is acting on P2X4 sensory neurons in response to noxious stimuli, we tested the needle assay in P2X4-sensory neuron deficient mice. Indeed, the noxious as well as innocuous mechanical responses were decreased in the P2X4-sensory neuron deficient mice, whereas the null responses increased (Figure 7). Finally, to determine whether P2X4 channels are specifically mediating these effects, we tested the P2X4-sensory neuron deficient mice in combination with apyrase injection; and found no additive effect of apyrase treatment and knockdown of P2X4 (Figure 7). These results suggest that ATP release from keratinocytes does not only mediate innocuous mechanical responses, but also noxious mechanical responses via P2X4 signaling on sensory neurons.

c) We have Discussed these new data in light of previous literature (including the Caterina and Albers studies) in the Discussion section: subsection “Keratinocyte activation elicits attending and “grooming” behaviors”; as well as subsection “Keratinocytes communication is not fiber type specific”, first paragraph.

Other points to address:

1) How do the authors explain the long latency (avg 46 seconds) to elicit behavioral responses following keratinocyte activation? Is keratinocyte-to-sensory neuron transmission a sufficient mediator of sensory neuron activation or is it permissive? Does the long latency for behavioral responses following light stimulation support the latter? Please discuss.

We agree with the reviewers that this is a perplexing delay. We do not have a definitive answer; however, two possibilities are as follows: A potential reason is that the ATP concentration released from keratinocytes in vivo increases over time in the tissue before the nerve terminal receives a sufficient ATP concentration to be activated. It is plausible that keratinocyte depolarization does not evoke the same amount of ATP release as mechanical stimulation, therefore it needs to build up in the skin before the sensory nerve terminal is activated.

We believe that ATP is a potentiator rather than an initiator of action potential firing in response to mechanical stimulation of the skin. ATP is not likely required for the initiation of the sensory neuron response to occur, because mechanical responses still do occur even in the presence of keratinocyte-inhibition, ATP hydrolysis, or deletion of sensory neuron P2X4. However, in each of these situations, the mechanical responses of the sensory neurons are muted. Thus, the most parsimonious explanation is that the ATP released from keratinocytes serves as a potentiator of neuronal responses.

We have added information into the discussion on ATP likely being a potentiator of mechanical responses in the subsection “Keratinocyte-sensory neuron signaling serves as an potentiator of touch transduction”.

2) Related to the above, the optogenic inhibition experiments shown in Figure 1 used light stimuli that were applied a minute before and during the mechanical stimulus. Why one minute before? Does inhibition only at the time of mechanical stimulation attenuate behavioral responses? And what is the effect of light stimulation in Arch-K14 mice on the physiological response properties, as measured in the apyrase experiments?

We thank the reviewers for raising this question. We have included this test in Figure 1—figure supplement 1.

We chose to apply the stimulus for 1 minute to make the evoked behavioral assays more convenient to perform as well as to be certain that all of the animals had at least a minimally sufficient amount of light exposure. That said, to specifically address this question we tested whether pre-light exposure was necessary by turning on the 590 nm light at the same time as the mechanical stimulus was delivered, to the best of our human abilities. To our relief, there were no differences in the inhibitory effects on behavior when light and stimuli were delivered simultaneously compared to when animals were pretreated with light for 1 minute before stimuli were delivered to the skin.

3) The absolute ATP concentrations are likely to be rough approximations and likely underestimates, given that both keratinocytes and sensory neurons are coated with ectonucleotidases. Please discuss in terms of the measured concentrations, compared to reported EC50 values for P2X receptors.

We agree that the ATP concentrations reported in Figure 3 are most likely rough approximations of the actual ATP concentrations that occur in the keratinocyte-sensory neuron interface upon mechanical stimulation, where nerve terminals are buried deep within the skin. As the traces in Figure 3 suggest, the ATP signal was short in duration and quickly degraded upon release. This is most likely due to the ectonucleotidases expressed on keratinocytes and sensory neurons (Zylka et al. 2008; Sowa, Taylor-Blake, et al. 2010; Sowa, Voss, et al. 2010; Beckenkamp et al. 2014) as the reviewers suggest. Furthermore, the ATP concentrations reported in Figure 3 µM) are calculated from the area under the curve of the traces (i.e. the area measured encompassed 60 seconds of the recording), and do not correspond to the peak value of ATP release. The P2X2 receptors used for the estimation of ATP concentrations in the cell sniffer assay have an EC_50_ value of 1400 nM (rat P2X2 in xenopus oocytes) (Jacobson et al. 2002), and the mechanical stimulation of keratinocytes was sufficient to cause inward currents in the P2X2-GFP^+^ HEK293 cells. The EC_50_ values of ATP for P2X receptors vary from low nanomolar to high μM concentrations. The concentrations of ATP released obtained from our experiments (5-35 µM) are sufficient to activate most P2X receptors including P2X4 (EC_50_ 500 nM) (Jacobson et al. 2002). While according to EC_50_ values P2X4 is one of the less potent homomeric P2X channels, the native physiological estimation is complicated by the fact that P2X receptors often exist in heteromeric confirmations (i.e. P2X2/3) and reports of the EC_50_ for P2X2/3 receptors are challenging to find. Thus, it is difficult to draw definite conclusions from these numbers. Further, although this is speculative, it is possible that ATP is focally released in high concentration pocket “domains” between sensory neurons and keratinocytes. This would lead to higher ATP concentrations in those localized signaling regions than the generalized ATP concentrations we measured in our assays and could lead to differential activation of P2X receptors on specific nerve terminals.

We have added text on this point to Discussion:

“The keratinocyte-released ATP must be acting through a specific receptor or set of receptors on sensory nerve terminals in order to convey the innocuous and noxious touch signal(s) to the spinal cord. […] Additionally, it is possible that ATP is focally released in high concentration pocket “domains” between the sensory neuron and keratinocyte cell membranes, and therefore, the ATP concentrations that occur in those localized signaling regions might be much higher than the generalized levels we measured in our assays.”

4) Related to major point 4 above, if ATP is indeed released from keratinocytes in response to innocuous tactile stimuli, then according to the authors' model a gentle touch would be expected to activate nociceptors and evoke pain because nociceptive afferents express P2X receptors and can be excited by ATP. This is obviously not the case under normal conditions. Please discuss.

We thank the reviewers for raising this concern, in order to address if keratinocytes mediate more than innocuous tactile sensations we tested animals in nociceptive assays and added those to the manuscript in Figure 1, Figure 4, Figure 5, Figure 5—figure supplement 1, Figure 7 and H.

Thanks to the careful critique from the reviewers of the manuscript, we also evaluated keratinocyte contributions to nociceptive responses. As described in major point 4b), 590 nm light activation of Archaerhodopsin significantly reduced the noxious as well as innocuous responses while concomitantly increasing null responses as compared to the 490 nm light control and to the Arch-K14Cre^-^ animals (Figure 1). Similarly, ATP hydrolysis and P2X4 knockdown decreased both innocuous and noxious responses while increasing the null responses (Figure 4 and Figure 7). In addition to that, our data shows that ATP is the major molecule released from keratinocytes in response to noxious touch, because we do not observe any additive effects with Arch inhibition and ATP hydrolysis (Figure 5). Further we do not see any additive effects in ATP hydrolysis and P2X4 genetic ablation model (Figure 7), thereby indicating that ATP released upon noxious touch is most likely acting on P2X4 on sensory neurons.

While our data shows that ATP is important in both innocuous and noxious touch responses, we also have several pieces of evidence that indicate that ATP release is concentration dependent according to intensity of mechanical stimuli (Figure 3 as well as Figure 4) as well as dependent on the light intensity (Figure 3). Therefore, we believe that lower intensity mechanical stimuli would elicit release of lower levels of ATP and activation of touch receptors, whereas higher intensity force would elicit greater ATP and activation of nociceptors. It is likely that a summation of EPSPs occurs at the sensory nerve terminal levels, therefore once more ATP binds to more receptors, nociceptive responses would ensue. Therefore, it is possible that lower levels of ATP release will not cause the same downstream effects on the “postsynaptic” sensory nerve terminals as higher levels of ATP binding to the sensory nerve terminals, where summation could occur.

5) The authors cite a paper showing that P2X4 receptor mRNA is expressed in most of sensory neurons. However, several previous studies showed that functional P2X receptors in DRG neurons are mainly P2X3, P2X2+3 receptors as determined by electrophysiological approaches. These P2X receptors are often co-localized with TRPV1 receptors in nociceptors. ATP-evoked currents mediated by P2X4 receptors have not been emphasized in those prior studies. Please discuss these points.

We have added two figure panels to Figure 6 to address P2X2 and P2X3 receptor involvement at baseline mechanotransduction.

We agree with the reviewers that P2X4 receptors to this point have not been investigated under baseline conditions. However, we show via pharmacological and genetic knockdown of P2X4 specifically in sensory neurons, that inhibition or knockdown of this ion channel leads to significant mechanical deficits in the animals at baseline. Furthermore, to this date P2X2 and P2X3 receptors also have not been shown to be involved in baseline sensation. To address the involvement of P2X2 and P2X3 subtypes, we performed behavioral experiments using NF 110, which at a lower concentration (500 nM) targets P2X3 receptors, and at a higher concentration (5 mM) targets both P2X2 and P2X3. We show that neither dose affects paw withdrawal thresholds or responses to a suprathreshold stimuli (Figure 6) after 60min and included the 30 and 120min data above under major concern 2a). Therefore, it is unlikely that these two receptor subtypes play a significant role in the baseline effects of mechanotransduction that we are investigating in the current study.

In other fields, P2 receptors have been shown to have complex response patterns, and rather than having distinct individual roles, P2 receptors can work in concert with both additive and inhibitory interactions (Xing et al. 2016). P2X2/3 receptors have been shown to be involved in injury conditions, where total ATP concentrations are likely higher than in acute mechanical stimulation of the non-injured skin. Therefore, it is possible that P2X4 and P2X2/3 receptors can play differential roles in naïve tissue versus injured tissue, and this may depend on the ATP concentrations present in pocket “domains” of the extracellular space in between keratinocytes and sensory neurons.

We added discussion material on this point as follows:

“Although there are a number of P2X channels that have been shown to be expressed by sensory neurons (Kobayashi et al. 2005), we chose to investigate P2X4 because of its equal expression on both light touch and nociceptive neurons (Kobayashi et al. 2005). […] However, purinergic signaling appears to be more multifaceted than would be expected by simply determining the probability of ATP binding via EC_50_ values because 1) receptors can also exist in heteromeric confirmations, and 2) P2 receptors have been shown to have complex response patterns, where rather than having distinct individual roles, different P2 receptors have been shown to work in concert through having both additive and inhibitory interactions (Xing et al. 2016).”